# Crosstalk between bone metastatic cancer cells and sensory nerves in bone metastatic progression

Sun H Park[1],* , Shunsuke Tsuzuki[1,2],* , Kelly F Contino[1],* , Jenna Ollodart[1],* , Matthew R Eber[1], Yang Yu[1], Laiton R Steele[1] , Hiroyuki Inaba[2], Yuko Kamata[3], Takahiro Kimura[2], Ilsa Coleman[4], Peter S Nelson[4], Enriqueta Muñoz-Islas[5], Juan Miguel Jiménez-Andrade[5], Thomas J Martin[6], Kimberly D Mackenzie[7], Jennifer R Stratton[7], Fang-Chi Hsu[8], Christopher M Peters[6], Yusuke Shiozawa[1]

**Although the role of peripheral nerves in cancer progression has been appreciated, little is known regarding cancer/sensory nerve crosstalk and its contribution to bone metastasis and associated pain. In this study, we revealed that the cancer/sensory nerve crosstalk plays a crucial role in bone metastatic progression. We found that (i) periosteal sensory nerves expressing calcitonin gene–related peptide (CGRP) are enriched in mice with bone metastasis; (ii) cancer patients with bone metastasis have elevated CGRP serum levels; (iii) bone metastatic patient tumor samples express elevated calcitonin receptor-like receptor (CRLR, a CGRP receptor component); (iv) higher CRLR levels in cancer patients are negatively correlated with recurrence-free survival; (v) CGRP induces cancer cell proliferation through the CRLR/p38/HSP27 pathway; and (vi) blocking sensory neuron–derived CGRP reduces cancer cell proliferation in vitro and bone metastatic progression in vivo. This suggests that CGRP-expressing sensory nerves are involved in bone metastatic progression and that the CGRP/CRLR axis may serve as a potential therapeutic target for bone metastasis.**

## Introduction

In recent years, the impact of the tumor microenvironment on cancer progression and metastasis has been increasingly explored. Cells within the tumor microenvironment, such as immune cells, endothelial cells, and fibroblasts, contribute to the suppression of immune response, development of tumor angiogenesis, induction of the epithelial–mesenchymal transition in cancer, and formation of the premetastatic niche (Neophytou et al, 2021; Soongsathitanon et al, 2021). More recently, peripheral nerves have also been identified as one of the critical components of the tumor microenvironment (Wang et al, 2021). Clinically, when cancer cells infiltrate into the perineural sheath (known as perineural invasion), the prognosis of cancer patients is thought to be poor (Chen et al, 2019). Preclinical studies using rodent models have demonstrated the role of peripheral nerves in cancer development and metastasis. Indeed, it was found that (i) autonomic nerves promote prostate cancer development and dissemination to the bone through the $\beta$2-and $\beta$3-adrenergic pathway (Magnon et al, 2013); (ii) cholinergic nerves induce gastric cancer tumorigenesis through the acetylcholine/nerve growth factor axis (Hayakawa et al, 2017); and (iii) activated sensory neurons enhance the proliferation of melanoma by stimulating immune cells (Keskinov et al, 2016). Furthermore, denervation of autonomic nerves significantly decreased tumor growth and dissemination. It has been shown that (i) ablation of sensory nerves using capsaicin decreases the development of pancreatic ductal adenocarcinoma (Saloman et al, 2016); (ii) pharmacological and surgical denervation of the autonomic nervous system induces suppression of gastric tumorigenesis (Zhao et al, 2014); and (iii) ablation of sympathetic nerve fibers through surgical or chemical intervention has been found to decrease early prostate tumor development, whereas pharmacologically inhibiting parasympathetic nerve fibers reduces prostate cancer dissemination to distant organs, including bone (Magnon et al, 2013).

Bone, one of the most common sites of cancer metastasis, is highly innervated by autonomic and sensory nerves. Interestingly, a recent study has demonstrated that the enhancement of perineural invasion in prostate cancer patient biopsies is associated with bone metastasis (Ciftci et al, 2015). Histochemical analyses of human and mouse bones have revealed that nerve fibers densely innervate the periosteum, bone marrow, and mineralized bone

[1]Department of Cancer Biology and Atrium Health Wake Forest Baptist Comprehensive Cancer Center, Wake Forest University School of Medicine, Winston-Salem, NC, USA   [2]Department of Urology, Jikei University School of Medicine, Tokyo, Japan   [3]Department of Oncology, Jikei University School of Medicine, Tokyo, Japan   [4]Division of Human Biology, Fred Hutchinson Cancer Center, Seattle, WA, USA   [5]Unidad Académica Multidisciplinaria Reynosa Aztlán, Universidad Autónoma de Tamaulipas, Reynosa, Mexico   [6]Department of Anesthesiology, Wake Forest University School of Medicine, Winston-Salem, NC, USA   [7]Teva Pharmaceuticals, Redwood City, CA, USA   [8]Department of Biostatistics and Data Science Wake Forest University School of Medicine, Winston-Salem, NC, USA

Correspondence: yshiozaw@wakehealth.edu
*Sun H Park, Shunsuke Tsuzuki, Kelly F Contino, and Jenna Ollodart are co-first authors

(Mach et al, 2002; Steverink et al, 2021). When cancer cells metastasize to the bone, these bone metastatic cancer cells activate and sensitize bone-innervating sensory nerves, resulting in cancer-induced bone pain (CIBP). A full 80% of patients with bone metastases have CIBP (Berruti et al, 2000), and it often presents as their first symptom. CIBP significantly impairs the quality of life (QOL) of cancer patients with bone metastases and has been suggested to be a negative indicator of survival. In the Tax 327 trial of advanced prostate cancer patients (Tannock et al, 2004), which compared Taxotere plus prednisone versus mitoxantrone plus prednisone regimens, men with reduced pain lived 6 mo longer than those with elevated pain (HR = 0.59; $P < 0.001$) (Berthold et al, 2008). Another analysis within the same study found (i) an association between pain and post-progression survival, and (ii) a chemotherapeutic benefit only in men without pain exacerbation (Armstrong et al, 2010). Similarly, the ALSYMPCA trial, which investigated the role of radium-223 in bone metastatic prostate cancer patients, found that decreased pain levels correlated with increased overall survival (Parker et al, 2013). Together, these studies provide evidence supporting CIBP's negative impact on bone metastatic cancer patient prognosis.

CIBP involves sensory nerve sprouting and the synthesis of neuropeptides (Jimenez-Andrade et al, 2010a). The calcitonin gene–related peptide (CGRP) is a 37–amino acid neuropeptide widely distributed in the peripheral and central nervous systems (Wimalawansa, 1996; Doods et al, 2007) and in sensory neurons (Jimenez-Andrade et al, 2010a; Park et al, 2017). CGRP-containing sensory neurons constitute most of the neurons that innervate bone (Jimenez-Andrade et al, 2010b), and their sprouting is associated with skeletal pain (Hong et al, 1993; Ghilardi et al, 2012; Chartier et al, 2014; Mantyh, 2014). Importantly, levels of CGRP are increased in the serum of patients with advanced prostate cancer, including patients with bone metastasis compared to those with low-grade prostate cancer (Suzuki et al, 2006, 2009). Furthermore, CGRP is elevated in osteoblastic prostate cancer biopsies and may be associated with the aberrant bone remodeling that presents in patients with bone metastasis (Larson et al, 2013). We previously reported that prostate cancer cells adopt similar bone homing mechanisms to hematopoietic stem cells to gain access to the bone marrow (Shiozawa et al, 2011). Sympathetic nerves are a major component of the microenvironment for hematopoietic stem cells (Katayama et al, 2006; Mendez-Ferrer et al, 2010) and are also involved in the metastatic progression of prostate cancer to bone (Magnon et al, 2013). These findings suggest that cancer-associated nerves stimulate bone metastatic progression. However, whether the interactions between bone metastatic cancer cells and sensory nerves, especially those expressing CGRP, regulate tumor growth within the bone remains unclear.

In this study, we found that (i) sensory nerves expressing CGRP are enriched in the periosteum of mice with bone metastasis; (ii) cancer patients with bone metastatic disease have elevated CGRP serum levels; (iii) tumor samples from patients with bone metastases express higher levels of a CGRP receptor, calcitonin receptor-like receptor (CRLR); (iv) CGRP induces proliferation of cancer cells through the CRLR/p38/HSP27 pathway; and (v) blocking CGRP, derived from sensory nerves, by monoclonal antibody against CGRP can reduce bone metastatic progression in vivo. Collectively, these data suggest that CGRP-expressing sensory nerves are involved in bone metastatic progression and that the CGRP/CRLR axis may serve as a potential therapeutic target for bone metastatic disease.

# Results

## The presence of cancer in bone is responsible for CIBP and sensitization of the central nervous system in mice

To test the interaction between bone metastatic cancer cells and sensory nerves, we first confirmed that bone metastatic tumor growth can induce CIBP. To do so, we established tumor in mouse bone marrow using an intrafemoral injection, which is a previously well-established approach to establish tumor within the marrow (Schwei et al, 1999). Briefly, luciferase-expressing DU145 cells were injected directly into the femurs of immunodeficient mice, and the injection site was plugged with bone cement to delay the spread of the tumor into the adjacent soft tissue. The DU145 cell line was selected over other human prostate cancer cell lines (i.e., PC-3) because of its higher propensity to induce pain in preliminary studies (data not shown). Thereafter, we found (i) tumor growth by bioluminescence imaging (Fig 1A and B); (ii) tumor burden in the marrow by histology (Fig 1C); (iii) increased spontaneous guarding behavior by a guarding measurement assay (Fig 1D); and (iv) decreased average time spent at optimal running velocity by a running wheel assay (Fig 1E) in tumor-burdened mice. Spontaneous guarding is a more direct measure of ongoing or spontaneous skeletal-related pain (Guedon et al, 2016; Majuta et al, 2017). The guarding assay is a well-established measure of pain behavior in rodent CIBP studies and is reversible by common analgesics including morphine and gabapentin (Luger et al, 2002; Peters et al, 2005). Running wheel assays measure movement-evoked pain behavior; mice and rats with inflammation of the hind paw demonstrate reduced levels of activity in the running wheel apparatus (Stevenson et al, 2011; Johnson, 2016; Whitehead et al, 2017). Although voluntary running wheel performance may reflect a functional measure of limb use rather than a purely pain-related outcome, it is an effective measurement of rodent QOL (Stevenson et al, 2011; Johnson, 2016; Whitehead et al, 2017). Mice are highly motivated to use the running wheel; thus, reductions in speed or time spent running may reflect depression or anxiety (Stevenson et al, 2011; Johnson, 2016; Whitehead et al, 2017). As reduced mobility and disruption of daily physical activities are a significant clinical consequence of bone metastatic cancer, the running wheel behavior assay serves as a potential preclinical QOL measure (Stevenson et al, 2011; Johnson, 2016; Whitehead et al, 2017). In addition, we observed an up-regulation of an astrocyte marker, glial fibrillary acidic protein (GFAP) expression in the ipsilateral spinal cord (Fig 1F and G), which corresponds to the site of central projections of sensory neurons that innervate the tumor-inoculated femur, suggesting that the presence of tumor in bone induces central sensitization, which is identified by activation of astrocytes in the central nervous system (Ishikawa et al, 2018; Li et al, 2019). Furthermore, when DU145 was subcutaneously injected into the flank of immunodeficient mice, an increase in tumor volume was

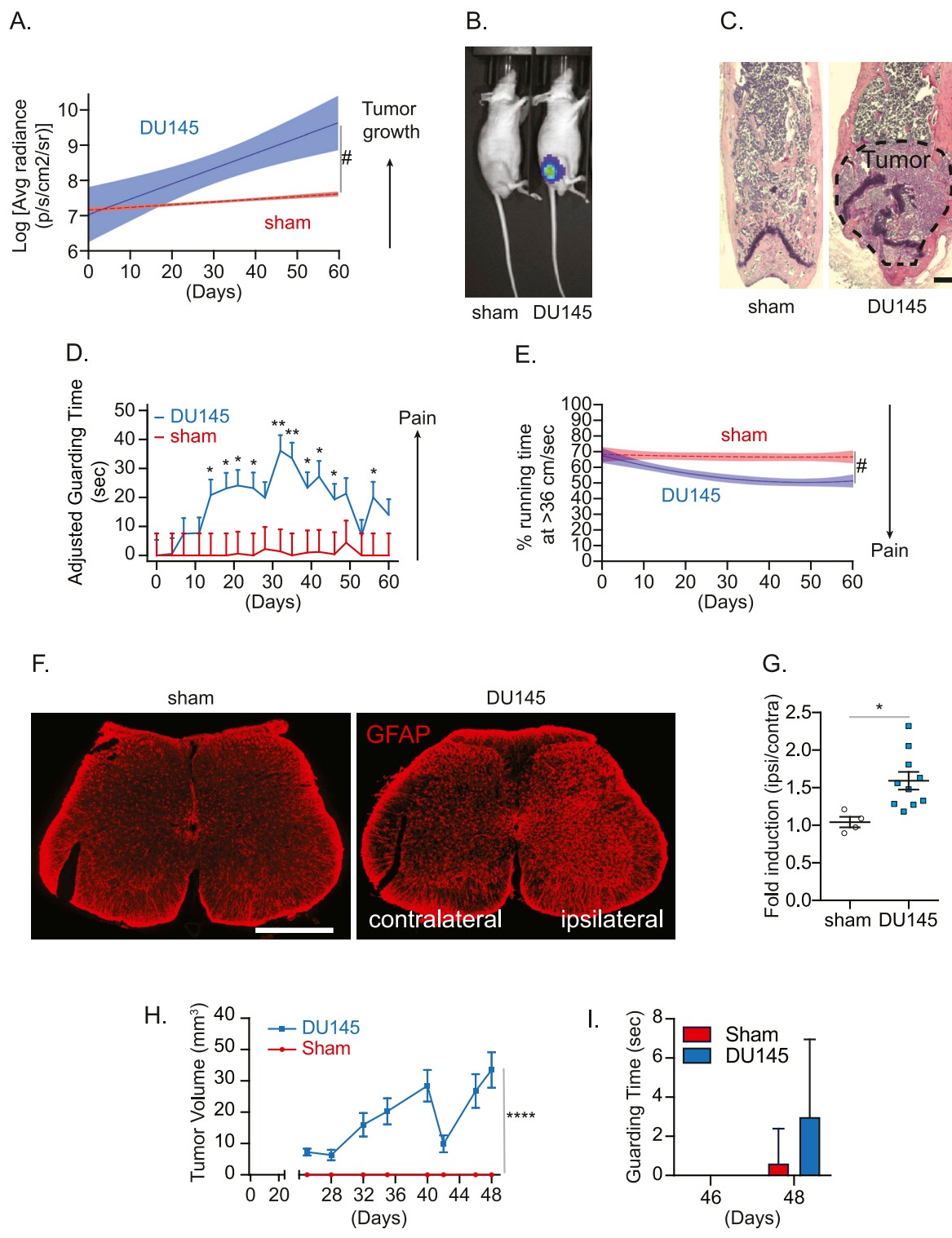

**Figure 1. In vivo cancer–induced bone pain model.**
Luciferase-expressing human prostate cancer cell line DU145 or conditioned medium (sham) was implanted directly into femurs of nude mice. **(A)** Bioluminescence imaging (BLI) was performed to measure tumor growth (n = 10/group). Data are the mean ± 95% confidence interval (CI). ($^{#}P \leq 0.05$ [time-by-group interaction] versus sham group [mixed-effects models]). **(B)** Representative bioluminescent images of sham- and luciferase-labeled DU145 cell–injected mice (week 8). **(C)** H&E staining of the femur of animals in (A, B). ×20. Bar = 500 $\mu$m. **(D)** Pain behaviors were measured by (D) guarding behavior measurement (mean ± SEM. **(E)** *$P \leq 0.05$ and **$P \leq 0.01$ versus sham group [mixed-effects models]) and (E) the running wheel assay (percent of time running at the optimal velocity of 36 cm/sec or greater) ($^{#}P \leq 0.05$ [time-by-group interaction] versus sham group [group-averaged growth curve models]). **(F)** Representative images of *glial fibrillary acidic protein*–immunostained spinal cord of animals in (A, B). ×10. Bar = 500 $\mu$m. **(F, G)** Quantification of (F). Data are the mean ± SEM. *$P \leq 0.05$ versus sham group (t test). **(H)** Tumor volume measured by a caliper. Data are the

observed, but there was no significant difference in guarding time (Fig 1H and I).

### The presence of cancer in bone induces the sprouting of CGRP-expressing sensory nerves, and CGRP can be released from bone metastatic cancer cell–associated sensory nerves

Next, we wanted to know whether bone metastatic cancer cells also influence the peripheral nervous system. We found more CGRP-expressing sensory nerve fibers in the periosteum of the ipsilateral tumor–bearing bones than the contralateral bones, whereas there were no changes in the sprouting of sensory nerve fibers in the bone marrow (Fig 2A and B; neurofilament 200 (NF200) was used as a marker for sensory nerves). Similarly, when primary mouse dorsal root ganglion (DRG) sensory nerves were treated with conditioned medium from cancer cell lines capable of metastasizing to the bone, sprouting of sensory nerve fibers significantly increased (Fig 2C and D), whereas conditioned medium from normal epithelial cells failed to induce their sprouting (Fig 2E).

It has been demonstrated that CGRP is released from the peripheral nerves into the extracellular space and bloodstream (Russell et al, 2014). Because we found more sensory nerves in periosteum of the tumor-bearing bones, our next question was to determine whether the resulting CGRP is secreted from these sprouted sensory nerves. In the animals that develop CIBP (A549-bearing mice) (Fig 3A), there were no changes in the levels of CGRP in the DRGs (Fig 3B) or the bone marrow (Fig 3C). When animals had bone metastasis and CIBP, elevated CGRP serum levels were observed (Fig 3D). Intriguingly, prostate cancer patients with bone metastatic disease had elevated CGRP plasma levels, compared to those without bone metastases (Fig 3E). Furthermore, CALCA (gene name of CGRP) mRNA expression was detected in DRG, whereas little to no CALCA mRNA expression was detected in cancer cells (Fig 3F).

### The higher expression of CRLR positively correlates with bone metastatic progression

Because we found (i) that the existence of cancer cells in the marrow induced the sprouting of CGRP-expressing sensory nerves and (ii) that the resulting CGRP might be in the bloodstream, we then wanted to determine whether CGRP derived from sensory nerves associated with bone metastatic cancer cells can influence bone metastatic progression. To do so, we investigated the function of the receptor for CGRP on cancer. CGRP receptor is a G protein–coupled receptor, which consists of CRLR and receptor activity–modifying proteins (RAMPs), which are necessary for the proper function of CRLR (McLatchie et al, 1998). There are three RAMPs, and CGRP has the highest affinity to the CRLR/RAMP1 complex, but it may also function through the CRLR/RAMP3 complex (Choksi et al, 2002).

Because prostate cancer patients develop bone metastasis at a later stage, we investigated the roles of CRLR and RAMPs in prostate cancer progression using a prostate cancer patient cohort (n = 390) in The Cancer Genome Atlas (Fig 4A). Interestingly, the higher expression of CALCRL (gene name of CRLR) is positively correlated with the higher Gleason score (Fig 4B), whereas RAMPs failed to show any positive correlation (Fig 4C–E). Even after adjusting for RAMP1 expression levels, age at diagnosis, and/or PSA levels, this correlation between CALCRL and the Gleason score still existed (Fig 4F). Moreover, the higher expression of CALCRL was negatively associated with their recurrence-free survival (Fig 4G). We also investigated the Gene Expression Omnibus (GEO) datasets. Similarly, the higher expression of CALCRL was positively associated with metastatic potential in prostate cancer patients (Fig 5A), whereas none of RAMPs nor CALCA were (Fig S1A–D). Intriguingly, samples from breast cancer patients with bone metastases expressed higher CALCRL levels, compared to those with brain, lung, or liver metastases (Fig 5B and C). However, none of the RAMPs nor CALCA were consistently expressed higher in the samples from breast cancer patients with bone metastases (Fig S1E–H and I–L). Similarly, CALCRL gene expression in samples from prostate cancer patients with metastases was higher than expression in benign prostate tissues and primary prostate cancer (Fig 5D). Importantly, when CALCRL gene expression levels were evaluated in samples from soft tissue metastases and bone metastases, the samples from bone metastases expressed higher CALCRL levels than those from soft tissue metastases (Fig 5D). Of note, normal bone-resident cells such as osteocytes also have some CALCRL expression.

Furthermore, the CRLR protein was more highly expressed in a tissue from a different cohort of patients with prostate cancer, compared with normal prostate tissues (Fig 6A and B). The higher CRLR expression levels positively correlated with the Gleason score (Fig 6C and D). Moreover, autopsy samples of bone marrow obtained from prostate cancer and lung cancer patients who died from bone metastases expressed higher levels of CRLR, compared with those obtained from non-bone metastatic prostate cancer and lung cancer patients that died because of other reasons (Fig 6E–G).

### CGRP induces proliferation of cancer cells that metastasize to bone through the CRLR/p38/HSP27 pathway

Because the data above suggest (i) that the presence of bone metastases is associated with CGRP release from sensory nerves and (ii) that its receptor CRLR is linked to bone metastatic progression, our next step was to determine whether CGRP can influence functional activities of cancer cells through CRLR. To do so, we first checked the expression levels of CRLR and RAMPs in cancer cell lines, which are capable of metastasizing to bone (human prostate cancer cell lines: LNCaP, PC-3, and DU145; human breast cancer cell lines: MCF-7, MDA MB-231, and ZR75-1; human lung cancer cell lines: A549 and SK-MES-1; and human prostate epithelial cell line: PWR-1E). The mRNA expression of CALCRL, RAMP1, and

---

mean ± SEM. ****$P$ ≤ 0.0001 versus sham group (two-way ANOVA). **(I)** Guarding behavior measurement. Data are the mean ± SEM; $P$ ≤ 0.05 is considered as statistically significant ($t$ test).

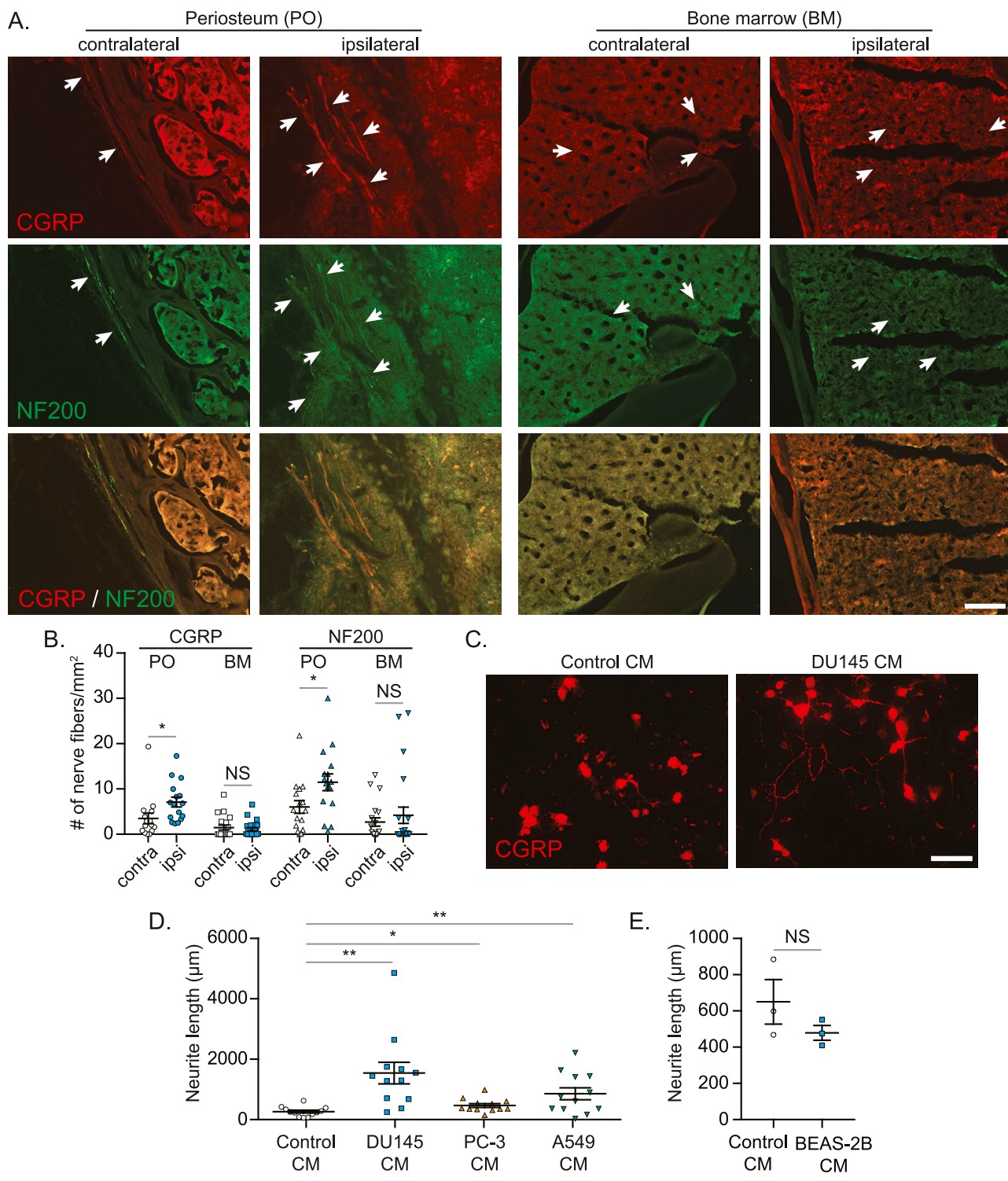

**Figure 2. Bone metastatic cancer cells increase CGRP-positive sensory nerve sprouting both in vitro and in vivo.**
**(A)** Representative images of *calcitonin gene–related peptide* (CGRP)–positive (red) and NF200-positive (green) sensory nerve sprouting in periosteum area bone marrow of animals in Fig 1A and B. 20x. Bar = 100 μm. White arrows indicate nerve fibers. **(A, B)** Quantification of nerves in mice shown in (A). Data are the mean ± SEM. *$P \le$ 0.05 versus contralateral bone (*t* test). **(C, D, E)** Primary dorsal root ganglion cells were treated with either control conditioned media (Control CM), or cancer cells (DU145, PC-3, or A549) or normal epithelial lung cell (BEAS-2B) conditioned media (DU145 CM, PC-3 CM, A549 CM, or BEAS-2B CM) for 48 h. **(C)** Representative images of CGRP-positive sensory nerve sprouting. 20x. Bar = 100 μm. White arrows indicate nerve fibers. **(D)** Quantification of the length of CGRP-positive sensory nerves exposed to Control CM, DU145 CM, PC-3 CM, or A549 CM. **(E)** Quantification of the length of CGRP-positive sensory nerves exposed to Control CM or BEAS-2B CM. Data are the mean ± SEM. *$P \le$ 0.05 and **$P \le$ 0.01 versus Control CM treatment (*t* test).

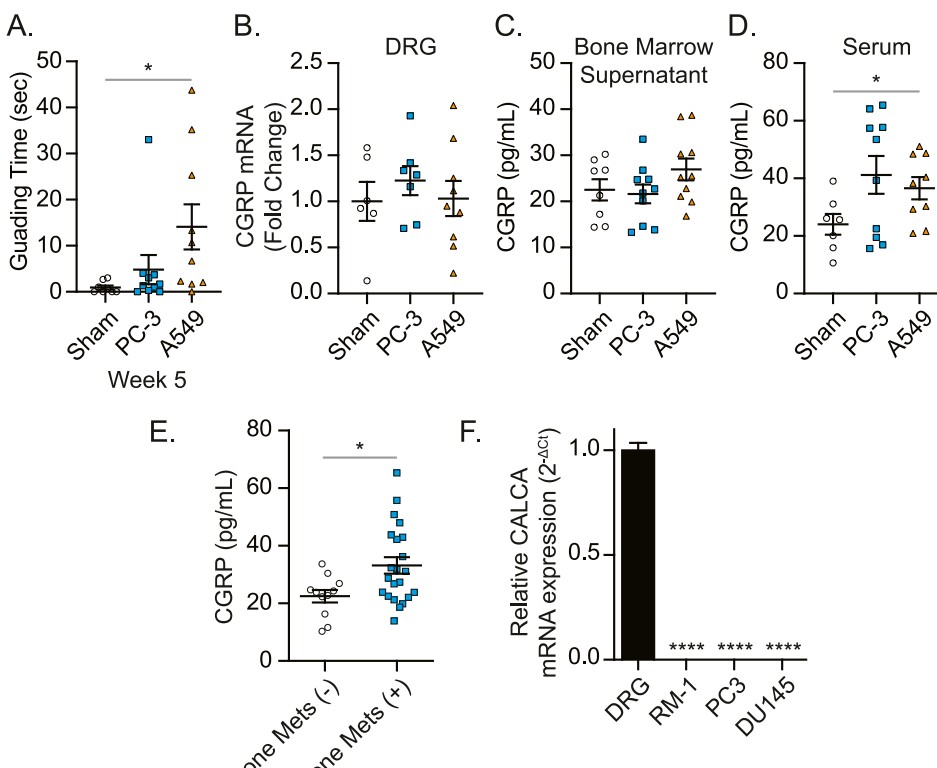

**Figure 3. Bone metastatic cancer significantly enhances CGRP levels in serum and plasma.** Human prostate cancer cell line PC-3, human lung cancer cell line A549, or conditioned medium (sham) was implanted directly into femurs of nude mice. **(A)** Pain behaviors were measured by guarding behavior measurement. Data are the mean ± SEM. *$P \leq 0.05$ versus sham group ($t$ test). **(B)** CGRP mRNA levels of ipsilateral dorsal root ganglia (L2-L4), **(C, D)** CGRP secretion in ipsilateral bone marrow, and (D) CGRP levels in serum among cancer-bearing mice and sham-injected mice were measured at week 5 after cancer inoculation. Data are the mean ± SEM. *$P \leq 0.05$ versus sham group ($t$ test). **(E)** Plasma *calcitonin gene–related peptide* levels of prostate cancer patients without ($n = 11$) and with ($n = 22$) bone metastasis. Data are the mean ± SEM. *$P \leq 0.05$ versus prostate cancer patients without bone metastasis ($t$ test). **(F)** CGRP mRNA expression of murine (RM-1) and human (PC3, DU145) prostate cancer cells and dorsal root ganglia using GAPDH as a reference gene and DRG set to 1. Data are the mean ± SEM. ****$P \leq 0.0001$ versus DRG (one-way ANOVA, Dunnett's multiple comparisons).

RAMP2 was detected in all the cells tested, whereas little to no RAMP3 mRNA expression was detected in all cell lines except for MCF-7 (Fig S2A–D). In addition, all the cells tested expressed CRLR and RAMP1 protein, whereas none of them expressed RAMP2 and RAMP3 protein (Fig S2E).

Thereafter, we tested the effects of CGRP on proliferation of cancer cell lines using MTT assays (Figs 7A and S3A), IncuCyte cell proliferation assays (Fig 7B–E), and doubling time counting analyses (Fig 7F). 2D cell culture models were used for these experiments as there is no in vitro model that recapitulates the dynamic micro-environment that is the bone. In short, CGRP induces faster pro-liferation in most of the metastatic cells tested. However, CGRP failed to alter the proliferation rate of cells from human prostate cancer cell line LNCaP and human breast cancer cell line MCF-7, which are known to have lower metastatic potential (Ravenna et al, 2014; Comşa et al, 2015; Sun et al, 2016). As expected, CGRP-mediated cell proliferation seen in Fig 7A was inhibited by a CGRP antagonist (Fig 7G and H). To validate whether CGRP derived from sensory nerves is responsible for the induction of proliferation of cancer cells, we also tested the effects of substance P, a neuropeptide known to be up-regulated along with CIBP (Lozano-Ondoua et al, 2013), on cancer cell proliferation. Contrary to CGRP, substance P did not enhance proliferation of cancer cell lines (Fig 7I). Interestingly, there was no difference in the substance P plasma levels between prostate cancer patients with bone metastatic disease and those without bone metastases (Fig 7J).

To further identify the molecular mechanisms of CGRP-mediated cancer cell proliferation, we performed the PathScan Intracellular Signaling Array, which tests the phosphorylation status of 16 well-known intracellular signaling molecules (Fig S3B) using LNCaP and MCF-7 cells as non-responders and DU145 and MDA MB-231 cells as responders, based on the results from Fig 8A. After incubating with CGRP for 8 h, p38, Stat1, p53, and HSP27 were identified as potential candidates downstream of the CGRP/CRLR axis, because these pathways were up-regulated in both responders and down-regulated in both non-responders (Figs 8A and S3C). We then verified the activation of p38 and its downstream protein HSP27 by Western blot in human cancer cell lines (Fig 8B) and murine prostate cancer cell line RM-1, although in RM-1, HSP27 was not activated (Fig S3C). However, the activation of Stat1 and p53 by CGRP was not detected (data not shown). As expected, the activation of p38 and HSP27 mediated by CGRP was inhibited by a selective p38 inhibitor (SB203580) (Fig 8B). More importantly, CGRP-mediated cell proliferation, seen in Fig 7A, was also inhibited by SB203580 (Fig 8C).

Interestingly, CGRP-mediated activation of these downstream targets was also inhibited by the CGRP receptor antagonist, CGRP8-37 (Fig 9A). Because we saw that this CGRP receptor antagonist can reduce these downstream targets and cell proliferation in vitro, we then investigated whether this phenomenon is recapitulated in vivo. To accomplish this, we treated RM-1–bearing mice with either vehicle or the CGRP receptor antagonist (CGRP 8–37) daily from days 1 to 18 (Fig 9B). Interestingly, we found no significant difference in bone remodeling, guarding time, or tumor growth (Fig 9C–E). Furthermore, we saw no change in the activation of p38 (Fig 9F and G).

A.

| | Mean ± Standard deviation |
|---|---|
| Age at diagnosis, year | 60.9 ± 6.9 |
| CALCRL | 8.0 ± 1.1 |
| RAMP1 | 11.4 ± 1.3 |
| RAMP2 | 7.9 ± 0.8 |
| RAMP3 | 8.3± 0.9 |
| PSA | 1.0 ± 0.4 (0.1) |
| Gleason score | 7.6 ± 1.0 |
| Overall survival, count (%) | 459 (98.5%) |
| Follow-up time for overall survival, day (median) | 943.0 ±739.6 (783.5) |
| Recurrence free survival, count (%) | 353 (90.5%) |
| Follow-up time for reccurrence free survival, day (median) | 939.0 ± 718.3 (766.0) |

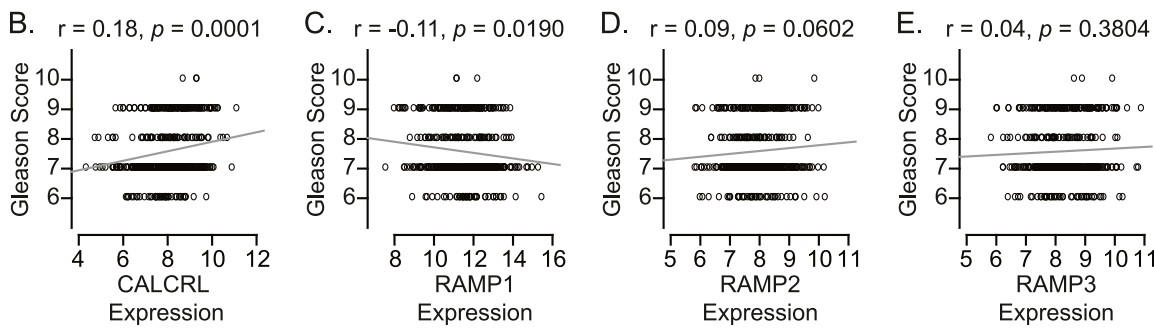

B. r = 0.18, $p$ = 0.0001
C. r = -0.11, $p$ = 0.0190
D. r = 0.09, $p$ = 0.0602
E. r = 0.04, $p$ = 0.3804

F.

| Variable | Comparison (Gleason score) | Model 1 | | Model 2 | | Model 3 | |
|---|---|---|---|---|---|---|---|
| | | Odds ratio (95% CI) | p-value | Odds ratio (95% CI) | p-value | Odds ratio (95% CI) | p-value |
| CALCRL | 7 vs. 6 | 1.2 (0.9, 1.6) | 0.1778 | 1.2 (0.9, 1.6) | 0.2429 | 1.3 (0.9, 1.8) | 0.1303 |
| | 8-10 vs. 6 | 1.5 (1.2, 2.0) | 0.0036 | 1.5 (1.1, 2.0) | 0.0143 | 1.6 (1.1, 2.2) | 0.0088 |
| RAMP1 | 7 vs. 6 | | | 0.9 (0.7, 1.2) | 0.3980 | 0.9 (0.7, 1.2) | 0.3532 |
| | 8-10 vs. 6 | | | 0.8 (0.6, 1.0) | 0.0877 | 0.8 (0.6, 1.1) | 0.1401 |
| Age at diagnosis | 7 vs. 6 | | | 1.0 (1.0, 1.1) | 0.0551 | 1.0 (1.0, 1.1) | 0.2134 |
| | 8-10 vs. 6 | | | 1.1 (1.0, 1.2) | 0.0004 | 1.1 (1.0, 1.1) | 0.0059 |
| PSA | 7 vs. 6 | | | | | 1.1 (0.8, 1.4) | 0.4578 |
| | 8-10 vs. 6 | | | | | 1.2 (0.9, 1.6) | 0.1955 |

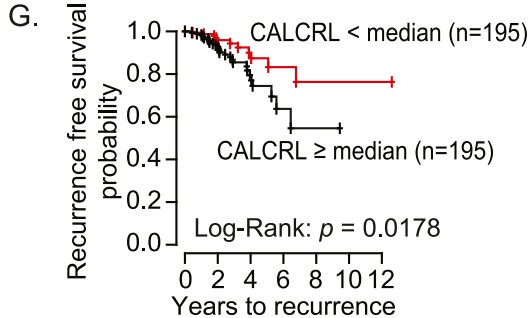

G. Log-Rank: $p$ = 0.0178

CALCRL < median (n=195)
CALCRL ≥ median (n=195)

### Blocking the binding of CGRP derived from sensory nerves to CRLR on cancer cells reduces bone metastatic progression

Although pharmacological antagonism of CGRP showed no change, we wanted to determine whether genetic manipulation of CGRP would mitigate bone metastatic progression and CIBP. To do so, we compared the bone metastatic progression between CGRP intact mice (referred to as CGRP Control mice) and CGRP global knockout mice (referred to as CGRP KO mice), in which the CGRP gene is linked with the GFP gene (Fig 10A and B). As expected, the DRG and spinal cord of CGRP Control mice expressed CGRP, whereas those of CGRP KO mice did not express CGRP immunoreactivity (Fig 10A and B). However, CGRP KO did not affect substance P immunoreactivity in the spinal cord of CGRP KO mice (Fig 10B). CGRP is known to be an important factor for bone formation and growth (Sample et al, 2011; Appelt et al, 2020; Kacena & White, 2020). Consistent with this point, we observed a reduction in baseline bone mineral density (BMD) in the trabeculae and cortical bone in femurs of CGRP KO mice compared with that of CGRP Control and WT mice (Figs 10C and S4A–D). Then, murine prostate cancer RM-1 cells were injected directly into the femurs of CGRP Control mice and CGRP KO mice. Surprisingly, there were no changes in tumor-induced bone remodeling (Fig 10D), spontaneous guarding behavior (Fig 10E), and tumor growth (Fig 10F).

We thought the global knockout of CGRP in sensory nerves induced some compensatory mechanisms to maintain bone metastatic progression. Therefore, we then decided to directly target the CGRP/CRLR axis with an antibody-based pharmacological approach. Because monoclonal antibodies targeting either the CGRP ligand or receptor are used clinically for migraine prevention (Tso & Goadsby, 2017; Bhakta et al, 2021), we treated A549-bearing immunodeficient mice with either murine monoclonal antibody against CGRP (anti-CGRP Ab) or isotype control antibody. However, there was no significant difference in tumor progression in the bone (Fig S5), which we hypothesized was due to the lack of an intact immune system. To test this, we then treated RM-1–bearing mice with either anti-CGRP Ab or isotype control antibody (Fig 11A). Although anti-CGRP Ab did not alter tumor-induced bone remodeling (Fig 11B) and spontaneous guarding behavior (Fig 11C), it significantly reduced tumor growth when compared to isotype control antibody (Fig 11D). In line with these findings, there was no change in tartrate-resistant acid phosphatase (TRAP)–positive osteoclasts between groups (Fig 11E and F). Although it did not reach statistical significance, anti-CGRP Ab treatment also reduces the activation of p38 (Fig 11G and H). Overall, directly targeting the CRLR/p38/HSP27 axis with anti-CGRP Ab may prove to be an alternative therapeutic avenue for treating bone metastasis. Future studies are needed to explore the use of p38 MAP kinase inhibitors or HSP27 inhibitors to inhibit bone metastatic progression.

We then sought to determine whether sensory nerves are responsible for CGRP-mediated bone metastatic progression. Interestingly, when cancer cells were co-cultured with DRGs, the proliferation of cancer cells was enhanced (Fig 12A). As seen previously, DRGs from CGRP KO mice failed to alter the proliferation of cancer cells (Fig 12B). However, when CGRP was blocked by anti-CGRP Ab in this co-culture, cancer cell proliferation was reduced (Fig 12C), although the growth of sensory nerves (Fig 12D) and cancer cell proliferation itself (Fig 12E and F) were not influenced.

## Discussion

Our data suggest that CGRP-expressing sensory nerves induce bone metastatic progression through CRLR (a component of CGRP receptors) expressed on bone metastatic cancer cells, by activating the p38/HSP27 pathway. Consistent with this notion, we also found (i) that cancer patients with bone metastasis had higher plasma CGRP levels compared to those without bone metastasis, (ii) that samples from bone metastatic tumors expressed higher levels of CRLR compared with those from other metastases, primary tumors, or benign tissues, and (iii) that CRLR expression in tumor negatively correlated with recurrence-free survival of cancer patients. Importantly, we also observed that anti-CGRP Ab treatments reduced bone metastatic progression in vivo and cancer cell proliferation in vitro by blocking CGRP derived from sensory nerves. These studies showed, for the first time to our knowledge, that the crosstalk between sensory nerves and bone metastatic cancer cells in the bone microenvironment impacts bone metastatic progression.

Owing to the advancements in prevention, screening, diagnosis, and treatment, the prognosis of cancer patients without metastasis has been substantially improved in recent years. However, once patients develop metastasis, their survival rate dramatically decreases. Although cancer cells may metastasize to any part of the body, bone is a major metastatic site for many solid cancers, including prostate cancer, breast cancer, and lung cancer (Svensson et al, 2017; Huang et al, 2020). Indeed, the median survival time of cancer patients after diagnosis with bone metastasis ranges from 68 to 377 mo depending on the primary tumor type (Svensson et al, 2017). Therefore, eradicating bone metastatic disease is critical if our goal is to prolong the lives of cancer patients. Unlike targeting primary tumor or other metastatic diseases, the treatment strategy for bone metastasis is quite unique because it is targeting the metastatic organ, bone. Current well-established treatments for bone metastasis are bisphosphonates and denosumab (anti-receptor activator of nuclear factor κ-B ligand [RANKL] antibody), which inhibit osteoclast activity. These agents have been effective in preventing the initial onset of skeletal-related events and bone

---

**Figure 4.  Higher CALCRL gene expression is associated with a higher Gleason score of prostate cancer.**
**(A)** Clinical information and *calcitonin gene–related peptide* receptor gene expression profiles (CALCRL, RAMP1, RAMP2, and RAMP3) of prostate cancer cohort (390 prostate cancer patients) obtained from The Cancer Genome Atlas database. **(B, C, D, E)** Correlation analyses between the Gleason score and the gene expression of (B) CALCRL, (C) RAMP1, (D) RAMP2, and (E) RAMP3. $P \leq 0.05$ is considered as statistically significant (Spearman's rank correlation coefficient). **(F)** Association analyses between the Gleason score, and CALCRL gene expression, RAMP1 gene expression, age at diagnosis, and PSA levels. $P \leq 0.05$ is considered as statistically significant (multinomial logistic regression models). **(G)** Kaplan–Meier curves for the association of CALCRL with recurrence-free survival. $P \leq 0.05$ is considered as statistically significant (log-rank test).

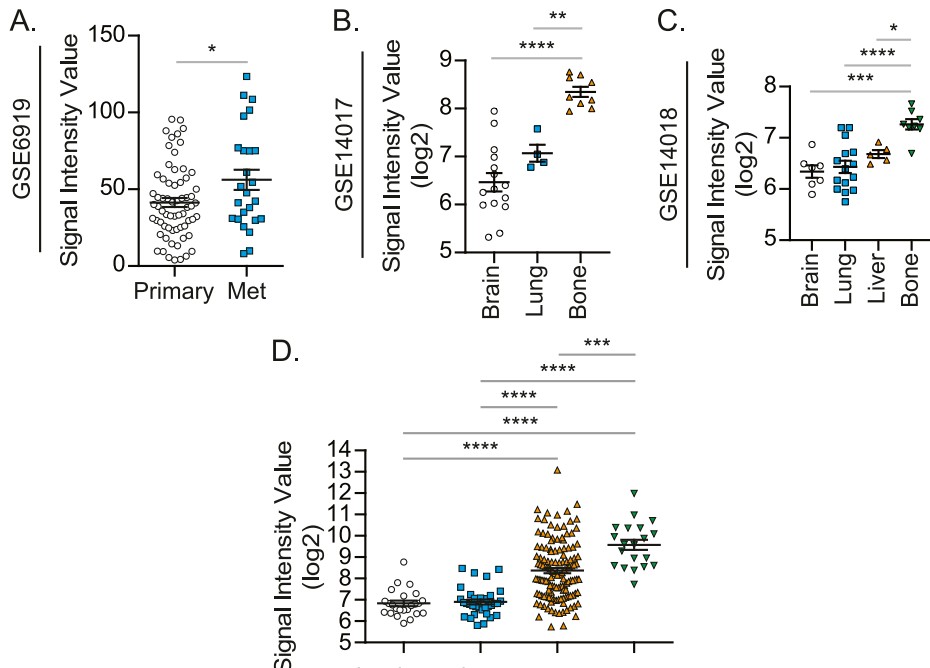

**Figure 5. CALCRL may be responsible for bone metastasis.**
**(A)** CALCRL gene expression in primary prostate cancer (Primary, n = 65) and metastatic prostate cancer (Met, n = 25) obtained from the GEO database (GSE6919). Significance versus primary prostate cancer (*t* test). **(B)** CALCRL gene expression in breast cancer with brain metastasis (Brain, n = 15), lung metastasis (Lung, n = 4), and bone metastasis (Bone, n = 10) obtained from the GEO database (GSE14017). Significance versus breast cancer with bone metastasis (one-way ANOVA, Dunnett's multiple comparisons). **(C)** CALCRL gene expression in breast cancer with brain metastasis (Brain, n = 7), lung metastasis (Lung, n = 16), liver metastasis (Liver, n = 5), and bone metastasis (Bone, n = 8) obtained from the GEO database (GSE14018). Significance versus breast cancer with bone metastasis (one-way ANOVA, Dunnett's multiple comparisons). **(D)** Expression of CALCRL in benign prostatic hyperplasia (n = 24), primary prostate cancer (n = 33), castration-resistant prostate cancer with soft tissue metastases (n = 129), and CRPC with bone metastases (n = 20) (one-way ANOVA, Tukey's multiple comparisons). Data are the mean ± SEM. *$P \le 0.05$, **$P \le 0.01$, ***$P \le 0.001$, and ****$P \le 0.0001$.

pain (Porta-Sales et al, 2017) but have failed to enhance overall survival of patients with bone metastasis by more than a few months (Scagliotti et al, 2012a, 2012b; Mollica et al, 2022). Despite the recent success of radium-223, which is known to target osteoblastic bone lesions, in prostate cancer patients with bone metastasis, radium-223 only enhances survival by a few months (Parker et al, 2013). As these treatments mainly target bone remodeling, new approaches that target alternative mechanisms are clearly warranted. In this study, we revealed for the first time that cancer-associated sensory nerves that innervate bone (and thus are directly involved in CIBP) play a crucial role in bone metastatic progression through the CGRP/CRLR axis. Although a recent study has demonstrated that exogenous CGRP supplement enhances bone metastatic progression in a rodent model, this study failed to elucidate the contribution of cancer-associated sensory nerves in the colonization and progression of metastatic cancer cells in the bone (Zhu et al, 2021). Our findings in vitro are consistent with these results where CGRP treatment causes enhanced proliferation. Importantly, we, for the first time, found that CGRP activates downstream signaling pathways such as phosphorylated p38 and HSP27 at varying times in four human cancer cell lines, although there is some minor variation in the time that CGRP induces activation. And the inhibition of p38 reversed the cancer cell proliferation mediated by CGRP. Despite these promising results in vitro, when using CGRP global KO mice, we observed no difference in bone metastatic progression. This result may be in part due to the compensatory mechanisms of a global CGRP KO; however, the reduction in bone metastatic progression seen with anti-CGRP Ab treatment could be the result of directly targeting CGRP pharmacologically. Because of this discrepancy, future studies are needed to study bone metastatic progression and CIBP using a conditionally inducible CGRP KO model to selectively KO CGRP in the sensory neurons of fully developed adult mice in order to reduce off-target effects observed with the global KO. In addition, the contribution of sensory nerves in communication with the tumor is a known mechanism of bone metastasis and CIBP. As such, understanding this crosstalk between sensory nerves and bone metastatic cancer cells will lay the foundation for the development of nerve/cancer-targeted therapies designed to minimize bone metastatic progression, which will ultimately allow for significant improvements in the care of affected patients.

Although, in our current study, anti-CGRP Ab treatment attenuated bone metastatic progression in vivo, this treatment strategy may not be effective as a monotherapy in the clinic, because, as mentioned above, current established treatments for bone metastasis using a single agent have only had limited success so far. Consistent with this notion, our CGRP receptor antagonist monotherapy also had limited efficacy in attenuating bone metastasis or CIBP. This difference between anti-CGRP Ab treatment and CGRP receptor antagonist may, in part, be due to the mechanism through which this axis is being targeted as anti-CGRP Ab binds directly with CGRP, whereas CGRP 8-37 is a receptor antagonist with high affinity for CRLR (Chou et al, 2022). However, CGRP does not exclusively act through CRLR, suggesting CGRP could still be playing a role in bone metastasis despite CGRP 8-37 treatment. Although the use of anti-CGRP Ab treatment as a monotherapy attenuated bone metastatic progression, this therapy may be more effective in combination with other anti-cancer treatments. Indeed, a variety of combinations of therapies (e.g., chemotherapy, radiation, hormone therapy) have recently been shown to modestly extend overall survival of cancer patients with bone metastasis (McCain, 2014). Therefore, combination approaches between therapies targeting the nerve/

**Figure 6. Calcitonin receptor–like receptor (CRLR) may be responsible for bone metastasis.**
**(A)** CRLR expression among prostate gland, prostate stroma, and prostate cancer areas in tissue microarray samples from prostate cancer patients. Bar = 100 μm. **(A, B)** Quantification of (A). Data are the mean ± SEM. **P ≤ 0.01 and ***P ≤ 0.001 versus prostate gland (one-way ANOVA, Tukey's multiple comparisons). **(C)** CRLR expression

cancer interaction and already existing treatments are particularly attractive strategies. The ideal combination tactic for bone metastasis needs to boost metastatic eradication while preserving bone health; however, finding this ideal combination has been challenging (Lee et al, 2011; Kim et al, 2019). This is in part due to the crosstalk between bone cells (e.g., osteoclasts, osteoblasts, and osteocytes) and bone metastatic cancer cells, which stimulates further bone metastatic progression through what is known as the "vicious cycle of bone metastasis" (Guise, 2002). Furthermore, these bone metastatic lesions present differently depending on the cancer type and thus need to be considered when developing treatment strategies. For example, prostate cancer bone metastasis typically presents with osteoblastic or osteogenic lesions, whereas lung and breast cancers have more osteolytic cancer-associated bone phenotypes (Regan et al, 2017; Wang et al, 2020). A recent trial of the combination of abiraterone (a new-generation androgen deprivation therapy) and radium-223 in patients with bone metastatic prostate cancer not only failed to improve skeletal event-free survival, but this combination strategy actually increased the frequency of bone fractures compared with placebo plus radium-223 (Smith et al, 2019). Consistent with this notion, androgen deprivation therapies are known to negatively affect bone health by enhancing osteolytic activities and reducing osteoblastic activities (Lee et al, 2011; Kim et al, 2019). The CGRP/CRLR axis is considered essential for bone formation (Sample et al, 2011; Appelt et al, 2020; Kacena & White, 2020). Interestingly, our CGRP global KO mice showed some degree of bone deficiency, suggesting that the blockade of CGRP may further suppress bone formation, resulting in increasing morbidity and mortality. Therefore, when choosing an effective combination strategy for bone metastases, agents that can enhance the efficacy of anti-CGRP Ab treatment while promoting bone formation through osteoblasts are necessary. Although inactive/immature osteoblasts reduce the sensitivity of cancer cells to existing chemotherapies (Sethi et al, 2011; Zheng et al, 2017), enhancing osteoblastic activity as a treatment strategy for bone metastatic disease has recently been appreciated (Toscani et al, 2018; Hesse et al, 2019). Thus, bone anabolic agents and/or agents inhibiting bone resorption (e.g., bisphosphonate, denosumab) can be a promising supplement to the blockade of CGRP for treating bone metastatic disease and its painful complications. One method that may prove useful for screening such potential combination therapies is 3D co-culture models designed to mimic the bone microenvironment. Indeed, a variety of 3D co-culture techniques have been used to test inhibitors for bone metastasis treatments and should be implemented in future studies (Marlow et al, 2013; Laranga et al, 2020). Nevertheless, it is important to consider the effect of heightened CGRP levels on different bone metastatic phenotypes when designing treatment strategies. However, more studies in this area are certainly warranted.

Known to be involved in CIBP is sensory nerve sprouting (Jimenez-Andrade et al, 2010a). Skeletal pain is associated with sensory nerve sprouting of CGRP-expressing neurons (Hong et al, 1993; Ghilardi et al, 2012; Chartier et al, 2014; Mantyh, 2014), and the levels of plasma CGRP directly correlate with pain intensity (Riesco et al, 2017; Schou et al, 2017). In addition, blocking the interaction between CGRP and CRLR can be effective for pain relief in migraine patients (Bigal et al, 2013; Tso & Goadsby, 2017). Moreover, CRLR antagonist treatments were found to reduce CIBP in a rodent model (Hansen et al, 2016). Therefore, we initially thought that targeting CGRP can attenuate CIBP. However, we did not observe the reduction in CIBP and sensory nerve sprouting by anti-CGRP Ab treatment, suggesting that CGRP alone is not the sole mechanism of CIBP. CIBP remains therapeutically challenging; it includes both spontaneous (ongoing) pain and breakthrough (movement-related) pain, which can present individually or in combination (Laird et al, 2011). Unless each component is appropriately treated, it cannot be managed. Analgesics that target the central nervous system (e.g., opioids, nonsteroidal anti-inflammatory drugs) are somewhat effective, but have severe side effects and are often addictive (Mercadante, 2001; Benyamin et al, 2008; Pergolizzi et al, 2008). External beam radiation, bisphosphonate, denosumab, and radium-223 can also reduce the onset of CIBP (Stopeck et al, 2010; Fizazi et al, 2011; Laird et al, 2011; Parker et al, 2013; Abou et al, 2015; Badrising et al, 2016; Vignani et al, 2016; De Felice et al, 2017), but they are primarily palliative and mainly target bone remodeling. Although better QOL may confer a survival benefit (Montazeri, 2009), most advanced cancer patients suffer from symptoms that negatively impact their QOL. Therefore, improving patients' QOL may prolong their overall survival. In the quest for effective cancer therapies, maintaining QOL is therefore as crucial as treating the tumor. So far, no treatment for CIBP targets the nerve/cancer interaction. In addition, our data suggest that CGRP may foster bone metastatic progression. Thus, a better understanding of the underlying mechanisms of the nerve/cancer interaction is also crucial to inform the development of safer and more effective therapies for both CIBP and bone metastasis. Ultimately, although further studies are clearly warranted, targeting the CGRP/CRLR axis and sensory nerve sprouting may provide the means to eradicate bone metastases and improve patients' QOL and survival.

In conclusion, our study is the first to significantly probe the mechanisms whereby the crosstalk between sensory nerves and bone metastatic cancer cells contribute to bone metastatic progression. Unveiling the molecular mechanisms of the cancer/nerve interaction may lead the way to improve overall survival and enhance QOL in patients with bone metastasis.

among different Gleason scores in tissue microarray samples from prostate cancer patients. Bar = 100 $\mu$m. **(C, D)** Quantification of (C). Data are the mean ± SEM. *$P \leq 0.05$ versus Gleason score (one-way ANOVA, Tukey's multiple comparisons). **(E)** CRLR expression in bone marrow autopsy samples from prostate cancer and lung cancer patients who died from bone metastases and other reasons. Bar = 100 $\mu$m. **(E, F)** Quantification of CRLR density between samples from cancer patients who died from bone metastases and other reasons in (E). **(E, G)** Quantification of CRLR density between bone marrow cells and cancer cells in (E). Data are the mean ± SEM. *$P \leq 0.05$ versus patients without bone metastasis ($t$ test).

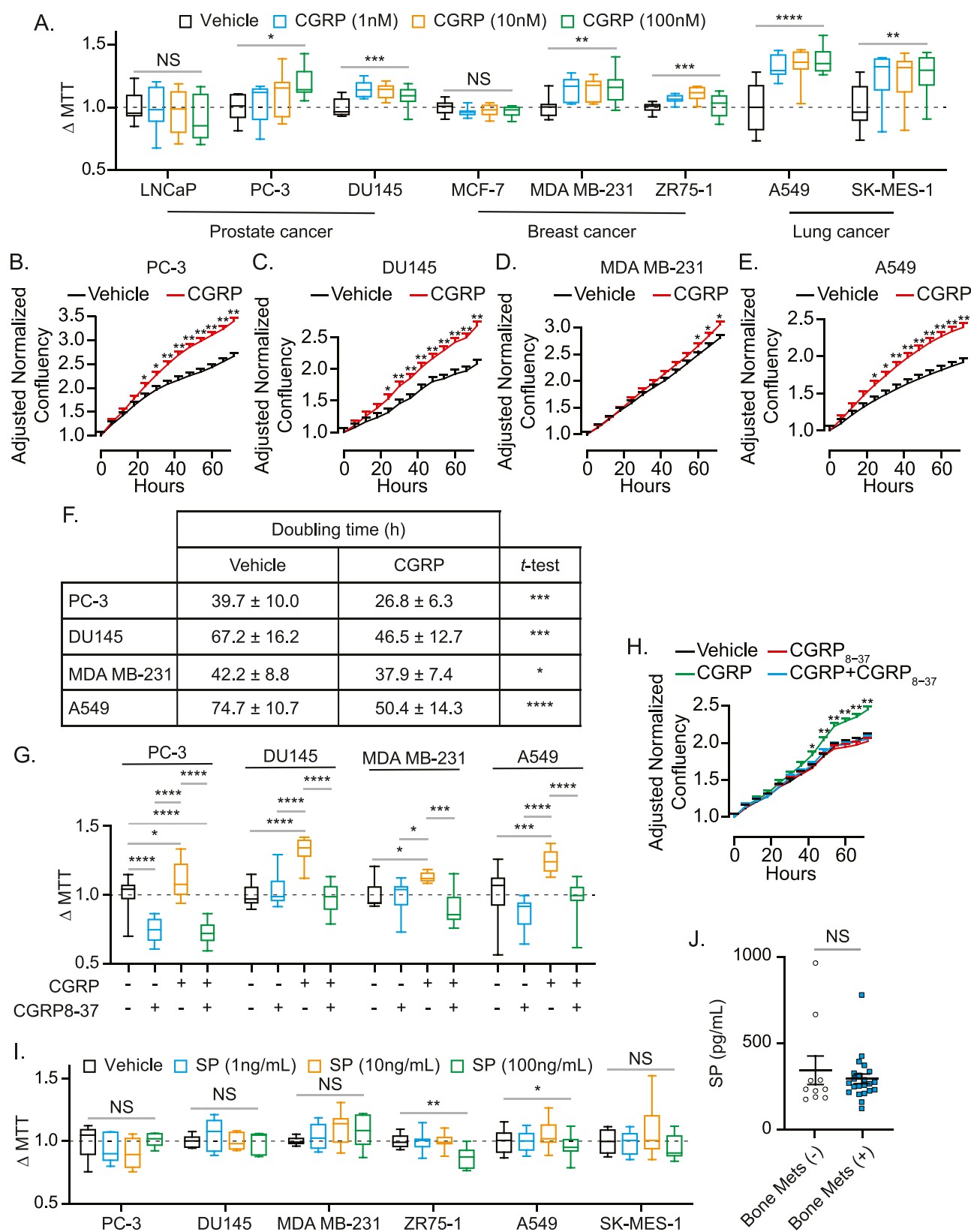

**Figure 7. CGRP induces proliferation of human cancer cell lines that metastasize to the bone through CRLR, but SP fails to induce their proliferation.**
**(A)** Box plots of cell viability (MTT) assays ± *calcitonin gene–related peptide* (CGRP) of human cancer cell lines (LNCaP, PC-3, DU145, MCF-7, ZR75-1, MDA MB-231, SK-MES-1, and A549) for 48 h. $P \leq 0.05$ is considered as statistically significant (one-way ANOVA). **(B, C, D, E)** IncuCyte ZOOM cell proliferation assays ± CGRP of human bone metastatic

# Materials and Methods

## Study approval

All human studies and all animal studies followed the Declaration of Helsinki and the Institutional Animal Care and Use Committee Guidelines, respectively. All animal studies were approved by the Institutional Animal Care and Use Committee at Wake Forest University School of Medicine (Protocol A15-056, A17-047, A18-026, A19-173, A19-192, A21-029). All human studies performed were approved by either (i) the Institutional Review Board at Jikei University School of Medicine (# 28-140[8383], 30-136[9157]; IRB) or (ii) the IRB at Fred Hutchinson Cancer Center (# IR6312 and IR7917; IRB). Informed consent was obtained from all subjects involved in the current study.

## Cell culture

Human prostate cancer cell lines PC-3 (Cat. #: CRL-1435), DU145 (Cat. #: HTB-81), and LNCaP (Cat. #: CRL-1740); human breast cancer cell lines MDA MB-231 (Cat. #: HTB-26), MCF-7 (Cat. #: HTB-22), and ZR75-1 (Cat. #: CRL-1500); human lung cancer cell lines A549 (Cat. #: CCL-185) and SK-MES-1 (Cat. #: HTB-58); murine prostate cancer cell line RM-1 (Cat. #: CRL-3310); and human lung epithelial cell line BEAS-2B (Cat. #: CRL-9609) were purchased from the American Type Culture Collection (ATCC). PC-3, DU145, LNCaP, ZR75-1, and A549 cells were maintained in Roswell Park Memorial Institute 1640 medium (Cat. #: 11-875-093; Thermo Fisher Scientific [Gibco]). MDA-MB-231, MCF7, SK-MES-1, and RM-1 cells were maintained in DMEM (Cat. #: 11-995-073; Thermo Fisher Scientific [Gibco]). BEAS-2B cells were maintained in BEGM (Cat. #: CC-3170; Lonza). All cultures were supplemented with 10% (V/V) FBS (Cat. #: 26-140-079; Thermo Fisher Scientific [Gibco]), 1% (V/V) penicillin–streptomycin (Cat. #: 15-140-163; Thermo Fisher Scientific [Gibco]), and 1% (V/V) L-glutamine (Cat. #: 25-030-164; Thermo Fisher Scientific [Gibco]). Cells were incubated at 37°C, 5% $CO_2$, and 100% humidity, and were routinely passaged when they were no more than 80% confluent.

Before the animal studies, some cancer cell lines (PC-3, DU145, A549, and RM-1) were transformed to stably express GFP and firefly luciferase by transduction with a lentivirus (Lenti-GF1-CMV-VSVG) generated by the University of Michigan Vector Core (Eber et al, 2021). The transduced cells were sorted for GFP-positive cells at the Wake Forest Baptist Comprehensive Cancer Center Flow Cytometry Shared Resource using Astrios EQ (Beckman Coulter), expanded, and frozen at low passage (<10).

## Conditioned medium collection

To collect the conditioned medium (CM) from control or cancer cells, $5 \times 10^5$ cells were seeded onto 10-cm dishes in complete growth medium. After 24 h, the medium was replaced with 10 ml of serum-free corresponding growth medium. For the control CM, 10 ml of serum-free DMEM or Roswell Park Memorial Institute medium was added to a 10-cm dish without adding any cancer cells. After 24 h of incubation at 37°C with 5% $CO_2$, the CM was collected, filtered through a 0.2-$\mu m$ syringe filter (Cat. #. CLS431222; MilliporeSigma) to remove any cell debris, and then stored at 4°C until use.

## Intrafemoral injection mouse model

Luciferase-expressing cancer cells were inoculated intrafemorally into mice. We used this previously well-established approach to establish tumor within the marrow (Schwei et al, 1999). Briefly, a one-cm incision was made in the skin of the right hindlimb (lateral side parallel to the femur) of mice to expose the muscle. The rectus femoris and vastus medialis muscles were separated using the line of connective tissue as a guide. Thereafter, the rectus femoris muscle and patella were moved to the medial side of the knee to expose the condyles of the femur. The connective tissue between the femur and the patella was cut, but the patellar tendon was not. A 27G needle was used to create the hole on the femur and then replaced with a C313I injector (Cat. #: C313I; Plastics One, Inc.) with Tygon tubing. After confirming that the injector was in the intra-medullary space by an X-ray (Faxitron Bioptics MultiFocus X-ray System), cancer cells (suspended in 5–10 $\mu l$ of Hanks' buffered saline solution [Cat. #: 14175103; Thermo Fisher Scientific (Gibco)]) were injected using a 10-$\mu l$ Hamilton syringe (Cat. #: 80300; Hamilton Company). The injection site was plugged with bone cement to delay the spread of the tumor into the adjacent soft tissue. The patella was then gently returned to its correct orientation, and to avoid possible patella displacement, muscles were secured back into position using a horizontal mattress suture technique and 7–0 absorbable sutures (Cat. #: 07-809-2011; Patterson Veterinary). Wound closure was then achieved with absorbable sutures. The same surgical procedure was used for sham animals except that the same amount of Hanks' buffered saline solution was injected instead of the cancer cells. Thereafter, tumor growth, CIBP behaviors, and bone remodeling were measured, as described below.

For the xenograft model, PC3, DU145, or A549 cells ($4 \times 10^4$ cells/10 $\mu l$) were inoculated into SCID Hairless Outbred mice (male, 4–6 wk old, Crl:SHO-$Prkdc^{scid}Hr^{hr}$, Cat. #: 474; Charles River Laboratories). For the syngeneic model, RM-1 cells ($1 \times 10^3$ cells/5 $\mu l$) were inoculated into CGRP homozygous (CGRP KO) (male, 4–6 wk old, B6.129P2[Cg]-$Calca^{tm1.1(EGFP/HBEGF)Mjz}$/Mmnc, Cat. #: 036773-UNC; Mutant Mouse Resource and Research Center at Jackson Laboratory), CGRP heterozygous (Control), or C57BL/6J (male, 4–6 wk old, Cat. #: 000664; Jackson Laboratory) mice. In some cases, mice were

cancer cell lines (B) PC-3, (C) DU145, (D) MDA MB-231, and (E) A549 for 72 h. Data are the mean ± SEM. Significance versus vehicle (mixed-effects models). **(F)** Doubling time ± CGRP of human bone metastatic cancer cell lines (PC-3, DU145, MDA MB-231, and A549) for 72 h. Data are the mean ± SEM. Significance versus vehicle ($t$ test). *$P ≤ 0.05$, **$P ≤ 0.01$, ***$P ≤ 0.001$, and ****$P ≤ 0.0001$. **(G)** Box plots of cell viability (MTT) assays ± *calcitonin gene–related peptide* (CGRP) ± CGRP$_{8-37}$ (*calcitonin receptor–like receptor* inhibitor) of human cancer cell lines (PC3, DU145, MDA MB-231, and A549) for 48 h. $P ≤ 0.05$ is considered as statistically significant (one-way ANOVA, Tukey's multiple comparisons). **(H)** IncuCyte ZOOM cell proliferation assays ± CGRP ± CGRP$_{8-37}$ of DU145 for 72 h. Data are the mean ± SEM. Significance versus vehicle (mixed-effects models). **(I)** Box plots of cell viability (MTT) assays ± substance P (SP) of human cancer cell lines (PC-3, DU145, ZR75-1, MDA MB-231, SK-MES-1, and A549) for 48 h. $P ≤ 0.05$ is considered as statistically significant (one-way ANOVA). **(J)** Plasma SP levels of prostate cancer patients without (n = 11) and with (n = 22) bone metastasis. Data are the mean ± SEM ($t$ test). *$P ≤ 0.05$, **$P ≤ 0.01$, ***$P ≤ 0.001$, and ****$P ≤ 0.0001$.

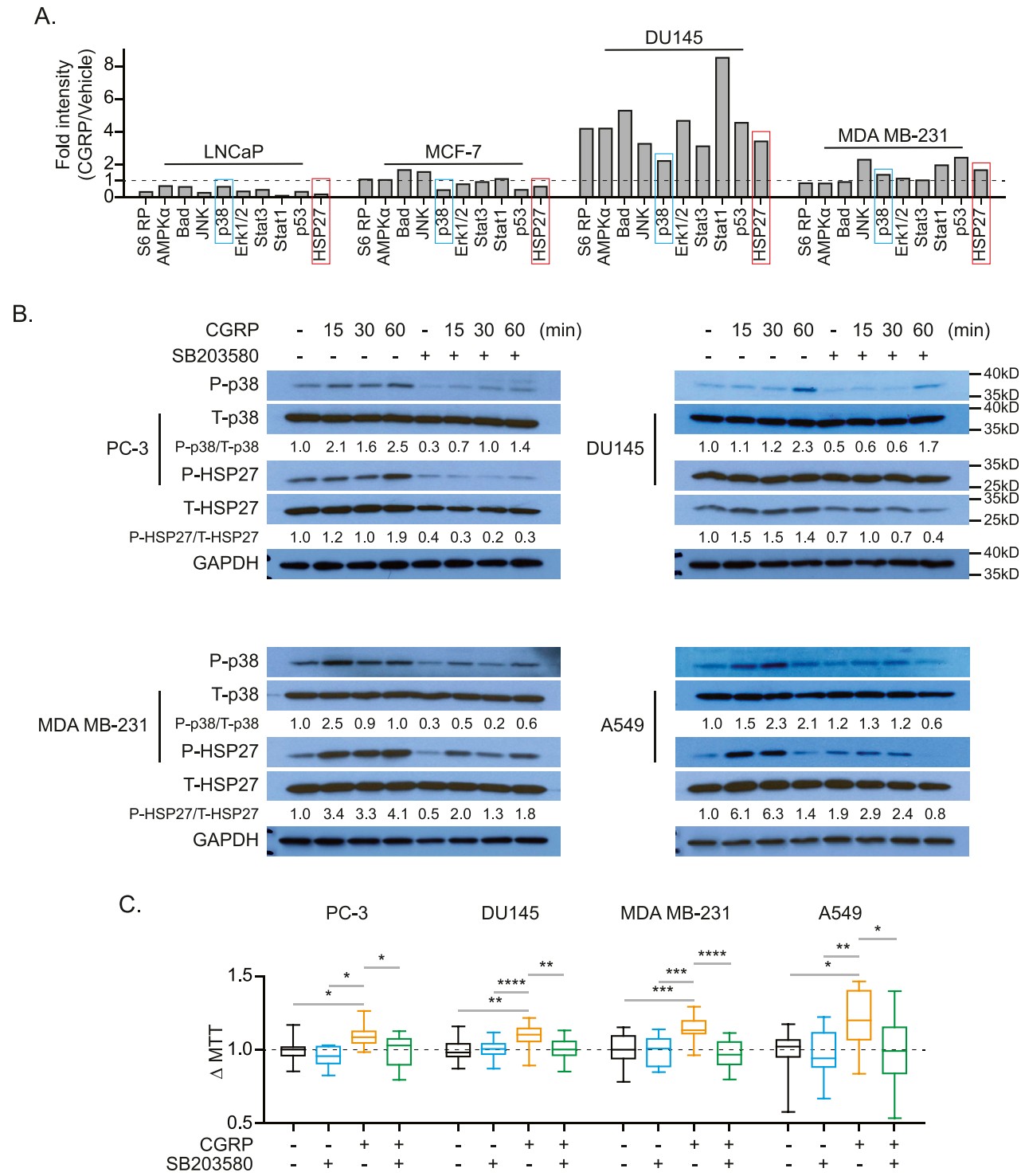

**Figure 8.** *Calcitonin gene–related peptide* (CGRP) induces proliferation of human cancer cell lines that metastasize to the bone by activating P38 and HSP27.
**(A)** Quantification of antibody-based cell pathway array data. DU145 and MDA-MB-231 cells (responders to CGRP), and LNCaP and MCF-7 cells (non-responders to CGRP) were exposed to CGRP for 60 min. **(B)** Representative Western blot of p38 and HSP27 phosphorylation ± CGRP ± SB203580 (p38 inhibitor) of human cancer cell lines (PC-3, DU145, MDA MB-231, and A549). GAPDH was used as a loading control. **(C)** Box plots of cell viability (MTT) assays ± CGRP ± SB203580 of human cancer cell lines (PC-3, DU145, MDA MB-231, and A549) for 48 h. $P \leq 0.05$ is considered as statistically significant (one-way ANOVA, Tukey's multiple comparisons). *$P \leq 0.05$, **$P \leq 0.01$, ***$P \leq 0.001$, ****$P \leq 0.0001$.

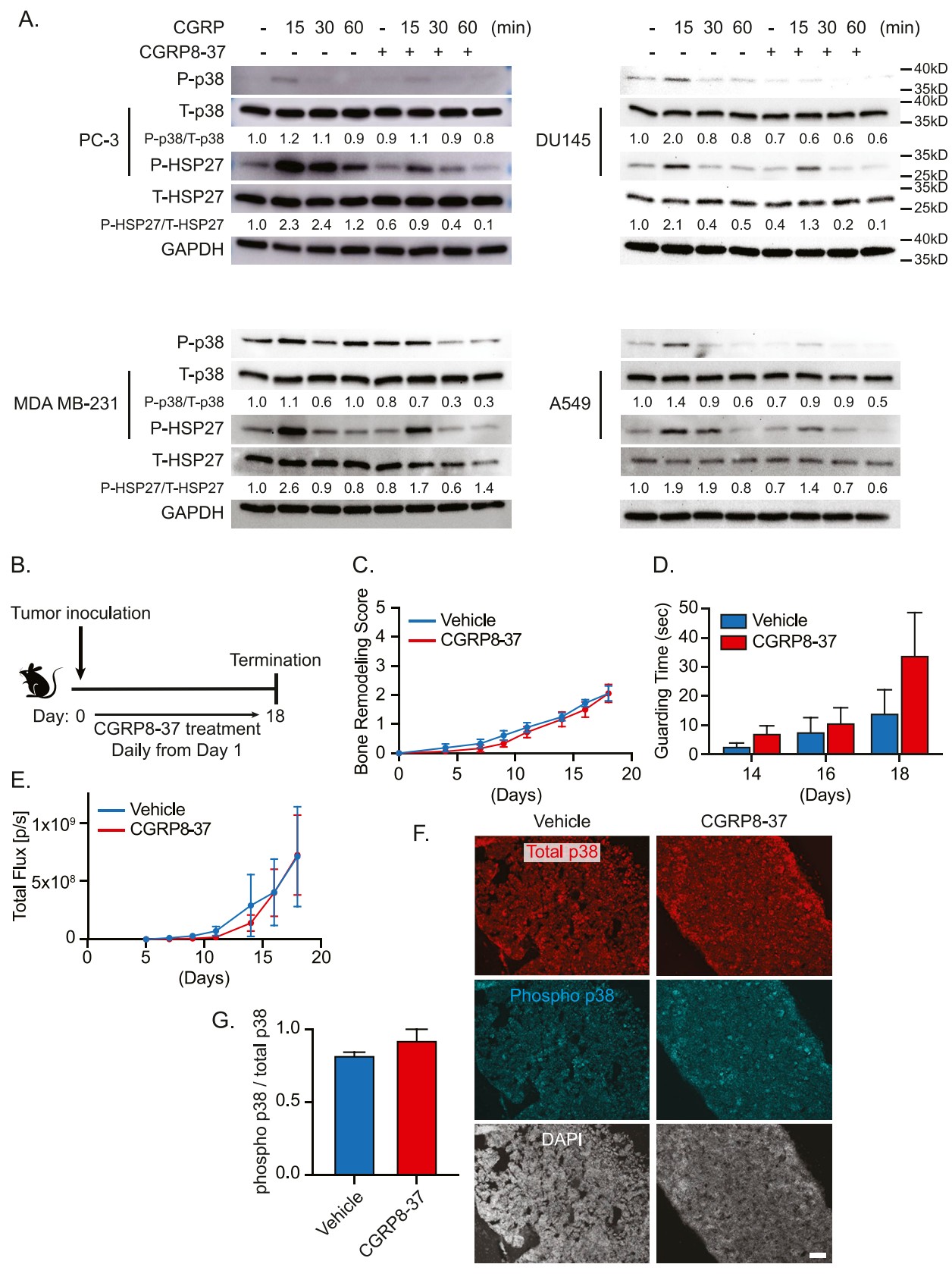

**Life Science Alliance**

treated with either mouse anti-CGRP monoclonal antibody or isotype control antibody (30 mg/kg, intraperitoneal injection; TEVA Pharmaceuticals) or vehicle or CGRP8-37 (100 µg/kg, intraperitoneal injection; R&D Systems).

## Measures of tumor growth

Bioluminescent images were obtained using IVIS Lumina LT Series III (PerkinElmer) through the Cell and Viral Vector Core Laboratory of the Wake Forest Baptist Comprehensive Cancer Center. Briefly, luciferase-expressing cancer cell–bearing mice were injected with luciferin (30 mg/ml, Cat. #: 122799; PerkinElmer) by intraperitoneal injections and ventral images were acquired 12 min post-injection under 2.5% isoflurane/air anesthesia. The total tumor burden of each animal was calculated using regions of interest (ROI) that encompassed the entire tumor area.

## Measures of CIBP behavior

Guarding behavior measurements were performed twice a week as previously described (Peters et al, 2004; Jimenez-Andrade et al, 2011). Tumor-bearing mice demonstrate a progressive increase in spontaneous pain behaviors, including guarding and periodic flinching of the inoculated limb. Mice were placed on nylon mesh platforms in clear plastic enclosures and acclimated for 30 min. Mice were observed for 5 min by an experimenter blinded to treatment groups.

The running wheel assay was performed five times a week using commercially available equipment (Med Associates, Inc.). Standard running wheel chambers contained a running wheel (18.54 cm diameter, 58.2 cm circumference) with a stainless-steel grid bar running surface located outside a 27.15 × 20.8 × 15.39 cm polycarbonate cage. Animals had free access to running wheels through a side opening in the cage. Each quarter turn of the running wheel operated a microswitch, and closures were recorded via a PC-compatible interface and computer (Med Associates, Inc.). Total microswitch closures (responses) and the time elapsed between microswitch closures (inter-response time [IRT]) were recorded using the MED-PC programming language. Each running wheel chamber was isolated in a PVC sound- and light-attenuating enclosure with a ventilator fan. For experiments involving running wheel assessment, mice were housed under a reverse light:dark cycle and running wheel sessions were conducted during the dark phase of the light:dark cycle on weekdays only. Mice were allowed free access individually to running wheels for 30-min sessions. Baseline measures for distance and optimal running rate were

obtained by averaging data from the last five sessions of this 3-wk period. In order to easily condition IRT data for distance traveled in running wheels and optimal running rate analysis, a Java program was developed to truncate and transfer the data daily from the MED-PC programming language into text files that would be easily importable into SAS. Responses were converted to distance traveled (14.55 cm per response), and IRTs were converted to speed (14.55 cm/IRT in seconds, cm/s). For optimal velocity determination, a paradigm was developed to find the running rates that best distinguished between sham and tumor-bearing animals. The optimal running rate was identified by quantifying the amount of time spent running at or above certain rates using lower and upper specification limits generated by SAS Proc Capability. The optimal velocity was generated using SAS specification limits.

## Measures of bone remodeling

Mice were X-rayed every week using a digital cabinet Faxitron Bioptics MultiFocus X-ray system. The X-ray images were provided to a blinded observer to perform longitudinal scoring of study endpoints using the following scales for assessing the extent of bone destruction (Eber et al, 2022): 0 = bones with no lesions; 1 = bones with one to three small pits of radiolucent lesions; 2 = bones with three to six small pits of radiolucent lesions; 3 = bones with obvious loss of medullary bone and erosion of cortical bone; 4 = bones with full-thickness unicortical bone loss; and 5 = bones with full-thickness bicortical bone loss and displaced skeletal fracture.

The trabecular bone was analyzed at the level of the distal femur and femoral neck, whereas the cortical bone was evaluated at distal femoral metaphysis using a micro-computed tomography ($\mu$CT) system (Skyscan 1272; Bruker). The scanning process was made at a 10 $\mu$m voxel size, and an X-ray power of 60 kVp and 166 $\mu$A with an integration time of 627 ms, according to the guidelines for $\mu$CT analysis of rodent bone structure (Bouxsein et al, 2010). Obtained images were reconstructed using NRecon software (Bruker). The trabecular ROI at distal femur metaphysis was evaluated by selecting 1 mm in the vertical axis, subsequent to 0.2 mm from the growth plate (reference point). For the cortical ROI analysis, the sample level was evaluated by selecting a band of 1 mm by 4 mm distal from the growth plate. For the femoral neck analysis, ROI was selected using a 0.5 mm$^2$ cylinder diameter, taking 1 mm of depth at 0.75 mm from the growth plate. The CT analyzer program (Bruker) was used to determine trabecular bone parameters; an automatic segmentation algorithm (CT analyzer) was applied to isolate the trabecular bone from the cortical bone. The parameters used for the trabecular bone were trabecular BMD, trabecular bone volume

**Figure 9. CGRP antagonism reduces p38 and HSP27 in vitro but does not attenuate bone metastatic progression nor p38 expression in vivo.**
**(A)** Representative Western blot of p38 and HSP27 phosphorylation ± CGRP ± CGRP8-37 (*CGRP receptor antagonist*) of human cancer cell lines (PC-3, DU145, MDA MB-231, and A549). GAPDH was used as a loading control. **(B)** Experimental schedule. Luciferase-expressing murine prostate cancer cell line RM-1 was implanted directly into femurs of C57BL/6 WT mice (n = 10/group). These mice were treated daily with either vehicle or CGRP8-37. **(C)** Bone remodeling was measured by X-ray. Data are the mean ± SEM. $P \leq 0.05$ is considered as statistically significant (mixed-effects model). **(D)** Pain behavior was measured by guarding behavior measurement. Data are the mean ± SEM. $P \leq 0.05$ is considered as statistically significant (*t* test). **(E)** Bone metastatic growth was measured by bioluminescence imaging. Data are the mean ± SEM. $P \leq 0.05$ is considered as statistically significant (mixed-effects model). **(B, C, D, E, F)** Representative images of total and phosphorylated p38–immunostained bone marrow of animals in (B, C, D, E). DAPI is used for nuclear staining. ×10. Bar = 100 $\mu$m. **(F, G)** Quantification of (F). Data are the mean ± SEM. $P \leq 0.05$ is considered as statistically significant (*t* test).

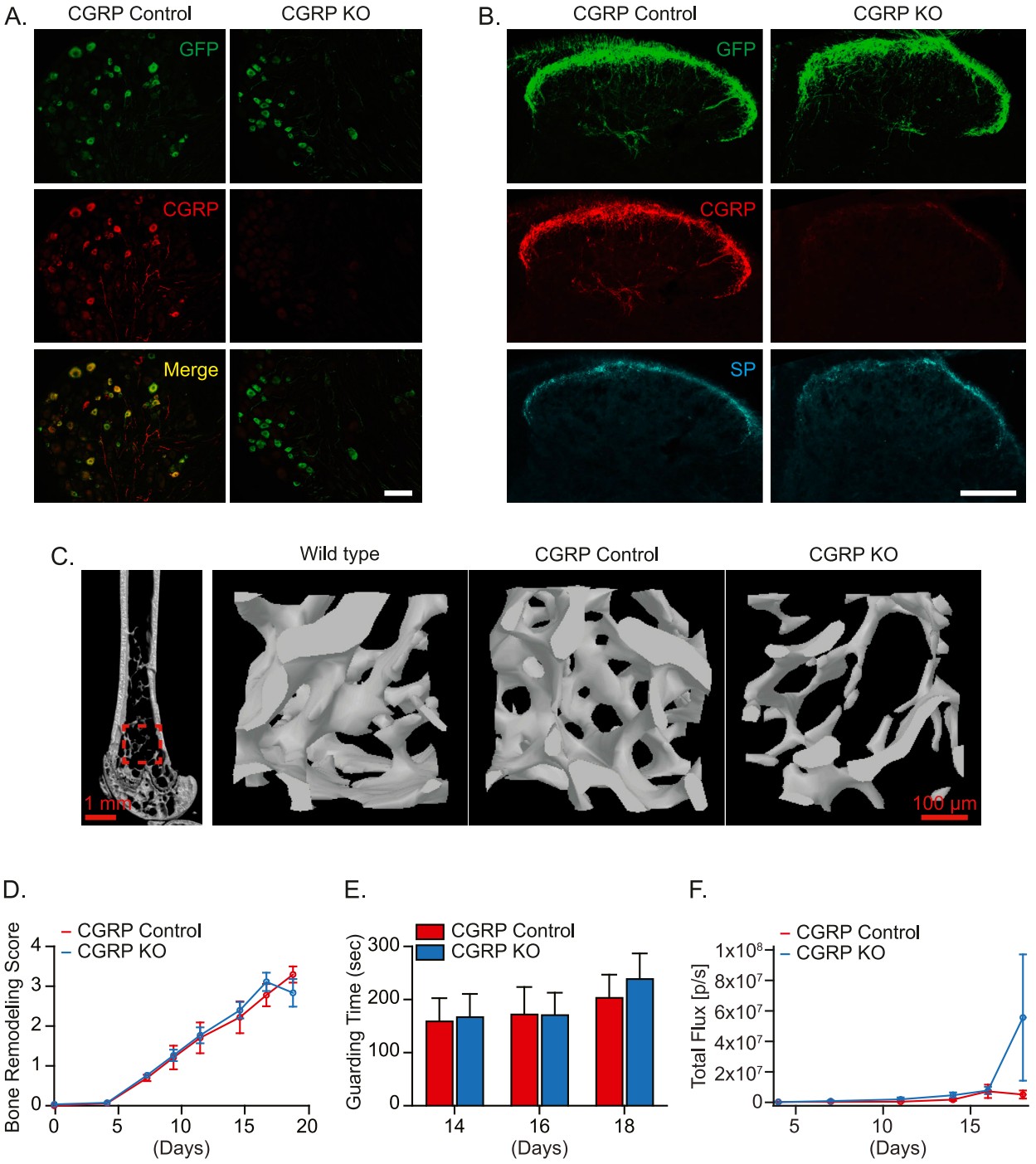

**Figure 10. Global *calcitonin gene–related peptide* (CGRP) KO fails to reduce bone metastatic progression.**
**(A, B, C)** Characterizations of CGRP KO mice. **(A)** Representative images of GFP and CGRP in the dorsal root ganglia of CGRP Control and CGRP KO mice. ×20. Bar = 100 $\mu$m.
**(B)** Representative images of GFP, CGRP, and substance P (SP) in the spinal cord of CGRP Control and CGRP KO mice. Bar = 100 $\mu$m. **(C)** Representative images of $\mu$CT scans of femurs from WT mice and CGRP Control and CGRP KO mice, including 2D slice image of the femur and 3D image of the distal trabecular area. **(D, E, F)** Luciferase-expressing murine prostate cancer cell line RM-1 was implanted directly into femurs of CGRP Control and CGRP KO mice (n = 10/group). **(D)** Bone remodeling was measured by X-ray. **(E)** Pain behavior was measured by guarding behavior measurement. **(F)** Bone metastatic growth was measured by bioluminescence imaging. Data are the mean ± SEM. $P \leq 0.05$ is considered as statistically significant (mixed-effects model for (D), $t$ test for (E), and mixed-effects model for (F)).

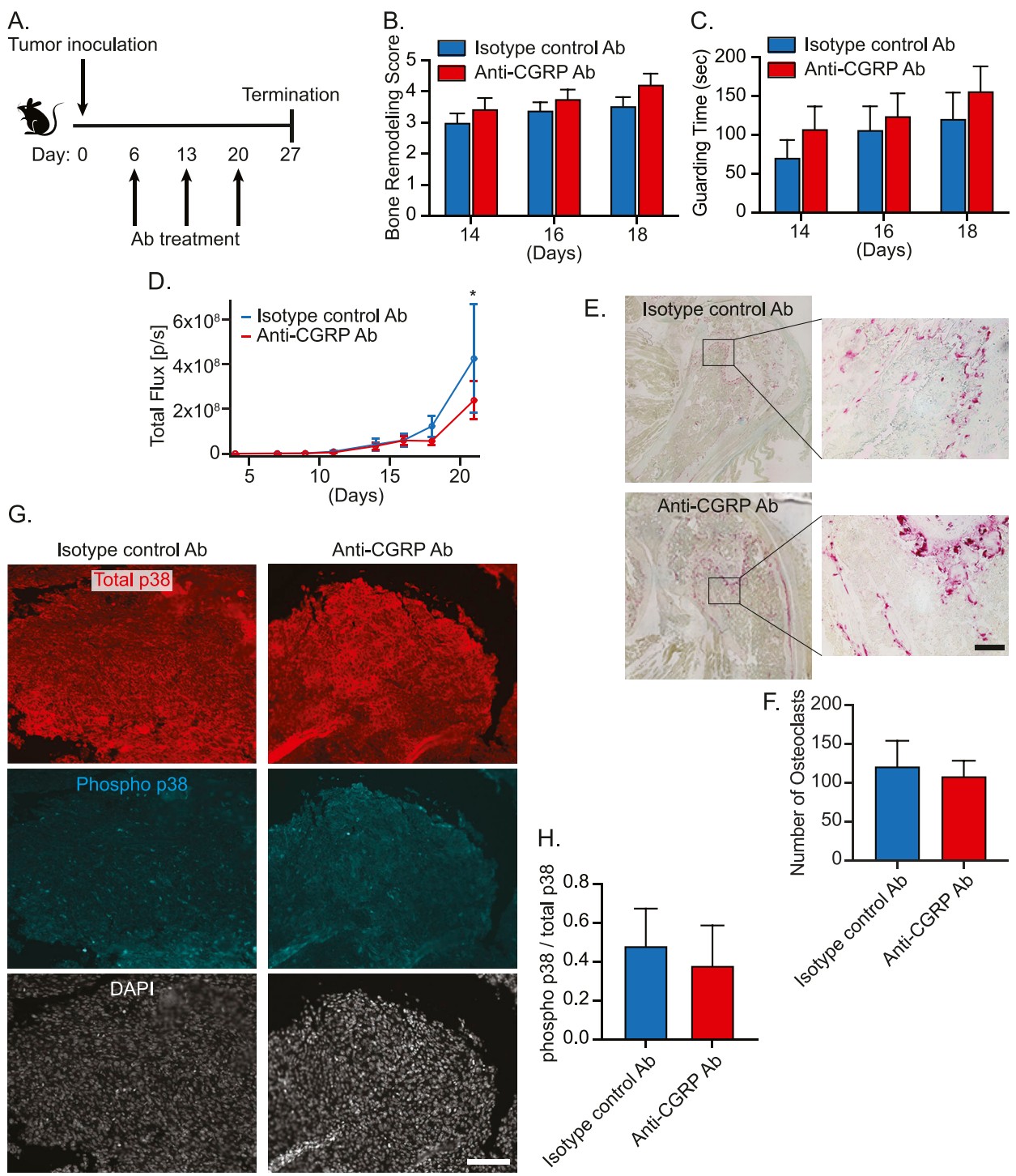

**Figure 11. Anti-*calcitonin gene–related peptide* (CGRP) monoclonal antibody treatment inhibits bone metastatic growth.**
**(A)** Experimental schedule. Luciferase-expressing murine prostate cancer cell line RM-1 was implanted directly into femurs of C57BL/6 WT mice (n = 20/group). These mice were treated intraperitoneally with either isotype control antibody (Ab) or anti-CGRP monoclonal Ab (30 mg/kg) at day 6, 13, or 20. **(B)** Bone remodeling was measured by X-ray. Data are the mean ± SEM. $P \leq 0.05$ is considered as statistically significant (*t* test). **(C)** Pain behavior was measured by guarding behavior measurement. Data are the mean ± SEM. $P \leq 0.05$ is considered as statistically significant (*t* test). **(D)** Bone metastatic growth was measured by bioluminescence imaging. Data are the mean ± SEM. *$P \leq 0.05$ versus isotype control antibody (Ab) (mixed-effects model). **(A, B, C, D, E)** Representative images of TRAP-positive osteoclasts in the bone marrow of animals in (A, B, C, D). ×20. Bar = 100 $\mu$m. (F) Quantification of (E). Data are the mean ± SEM. $P \leq 0.05$ is considered as statistically significant (*t* test). **(A, B, C, D, G)** Representative images of total and phosphorylated p38–immunostained bone marrow of animals in (A, B, C, D). DAPI is used for nuclear staining. ×10. Bar = 100 $\mu$m. **(G, H)** Quantification of (G). Data are the mean ± SEM. $P \leq 0.05$ is considered as statistically significant (*t* test).

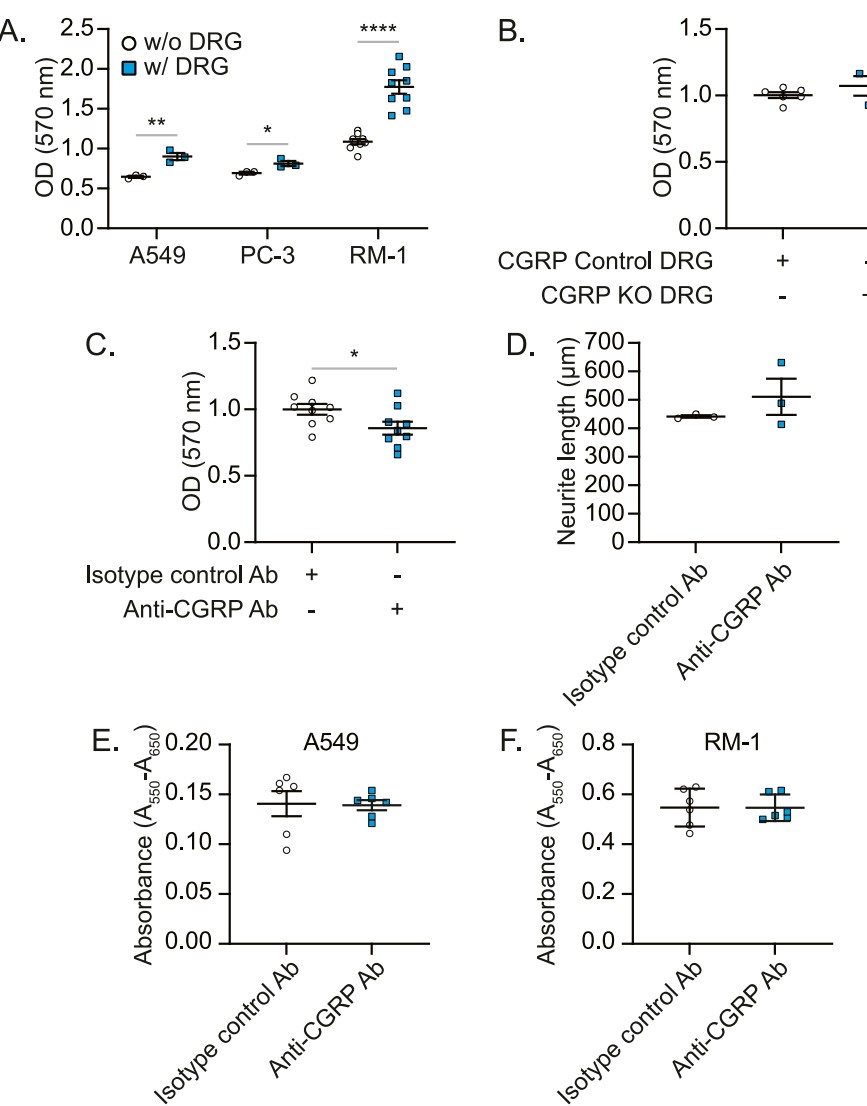

**Figure 12. Anti-*calcitonin gene–related peptide* (CGRP) monoclonal antibody treatment inhibits the proliferation of cancer cells mediated by sensory nerves, but not cancer cells alone.**

**(A, B, C, D)** Cancer cells and murine primary dorsal root ganglion (DRG) sensory neurons were co-cultured for 72 h. The numbers of cancer cells were measured using crystal violet staining at the termination of experiment. **(A)** Co-culture between cancer cells (A549, PC-3, and RM-1) and murine primary DRG sensory neurons from C57BL/6 WT mice. **(B)** Co-culture between cancer cells (A549) and murine primary DRG sensory neurons from CGRP Control and CGRP KO mice. **(C)** Co-culture between cancer cells (A549) and murine primary DRG sensory neurons from C57BL/6 WT mice treated with either isotype control antibody (Ab) or anti-CGRP monoclonal Ab. **(D)** Quantification of the length of murine primary DRG sensory neurons exposed to isotype control Ab or anti-CGRP monoclonal Ab for 48 h. Data are the mean ± SEM. $P \leq 0.05$ is considered as statistically significant ($t$ test). *$P \leq 0.05$, **$P \leq 0.01$, and ****$P \leq 0.0001$. **(E, F)** MTT proliferation of isotype control and anti-CGRP Ab-treated (E) A549 and (F) RM-1. Data are the mean ± SEM. $P \leq 0.05$ is considered as statistically significant ($t$ test).

rate (BV/TV), trabecular thickness (Tb.Th), trabecular number (Tb.N), and trabecular separation (Tb.Sp). Cortical bone parameters in 3D were cortical BMD, cortical thickness (Ct.Th), and 2D cross-sectional cortical bone area (Ct.Ar). Finally, hydroxyapatite calibration phantoms (250 and 750 mg/cm$^3$) were used to calibrate trabecular and cortical BMD values.

## Subcutaneous injection mouse model

Luciferase-expressing DU145 human prostate cancer cells were inoculated subcutaneously into the right flank of immunodeficient mice. Tumor growth was measured via a caliper.

## Animal tissue processing

At the termination of animal experiments, mice were perfused with 4% PFA. Spinal cords (L2-L4 region), DRGs (L2-L4), and femurs were collected, and these dissected tissues were prepared for immunohistochemistry and immunofluorescence (IF). In some cases, fresh serum, bone marrow supernatant, and DRGs were collected.

For frozen section preparation, spinal cords, DRGs, and femurs were post-fixed with 4% PFA for 12 h at 4°C, and then cryoprotected with 30% sucrose (Cat. #: S5-3; Thermo Fisher Scientific [Fisher Chemical]) solution in PBS (Cat. #: MT20031CV; Thermo Fisher Scientific [Gibco]) for 48 h at 4°C. Spinal cords and DRGs were embedded in Tissue-Plus optimum cutting temperature (Cat. #: 23-730-571; Thermo Fisher Scientific [Scigen Tissue-Plus]) compound, and frozen using dry ice. Femurs were decalcified with 10% EDTA (Cat. #: 327200025; Thermo Fisher Scientific [Fisher Chemical]) in pH 7.3 PBS for 14 d (the decalcification solution was replaced on day 8) before embedding in Tissue-Plus optimum cutting temperature compound and freezing. Frozen spinal cords, DRGs, and femurs were thaw-mounted on Superfrost non-adhesion glass slides (Cat. #: 12-550-143; Thermo Fisher Scientific) at 40 $\mu$m thickness, Superfrost Plus glass slides at 16 $\mu$m

thickness, and Superfrost Plus glass slides (Cat. #: 12-550-16; Thermo Fisher Scientific) at 20 $\mu$m thickness, respectively.

For paraffin section preparation, the dissected femurs were post-fixed with 4% PFA for 6 h at 4°C, and then decalcified with 10% EDTA, in pH 7.3 PBS for 14 d (the decalcification solution was replaced on day 8) at 4°C. Thereafter, femurs were embedded in paraffin and then cut into 5-$\mu$m-thick sections.

In some experiments, fresh serum, bone marrow supernatant, and DRGs were collected without perfusion. To collect bone marrow supernatant, tibias and femurs were flushed with 500 $\mu$l PBS, the resultant fluid was centrifuged, and the clear supernatant was collected.

### Nerve sprouting assay

Nerve sprouting assays were performed as previously described (Park et al, 2021). Briefly, a single-cell suspension of murine primary DRG sensory nerve cells was obtained from lumber DRG (L2-L4) of male C57BL/6 mice (8–12 wk of age) using enzymatic digestion and density gradient centrifugation. Then, 500–1,000 cells of DRGs in 30 $\mu$l of warm neuronal growth (NG) medium (Neurobasal-A [Cat. #: 21-103-049; Thermo Fisher Scientific (Gibco)], 1% N2 [Cat. #: 17502048; Thermo Fisher Scientific (Gibco)], 2% B-27 [Cat. #: 17-504-044; Thermo Fisher Scientific (Gibco)], 2 mM L-glutamine, 1% penicillin–streptomycin, and 0.4% glucose [MilliporeSigma]) were seeded onto the center of 14-mm round coverslips (Cat. #: P12G-1.5-14-F; MatTek Corp.), precoated with poly-D-lysine (50 $\mu$g/ml, overnight at 4°C, Cat. #: A3890401; Thermo Fisher Scientific [Gibco]) and laminin (20 $\mu$g/ml, 1 h at 37°C, Cat. #: CB-40232; Thermo Fisher Scientific [Corning]), in a 24-well plate. After 1–2 h, 1 ml of warm NG medium was gently added to the sides of wells and the cells were maintained at 37°C with 5% CO$_2$. After 48 h of primary DRG neuronal culture establishment, half (500 $\mu$l) of NG medium was replaced with 500 $\mu$l of either control or cancer cell–derived CM, and then, the DRG cells were incubated for another 48–72 h. In some cases, the DRG cells were treated with cancer CM, and either isotype or mouse anti-CGRP monoclonal antibody (100 nM; TEVA Pharmaceuticals). At the termination of the experiments, the DRG cells were fixed in 500 $\mu$l of 4% PFA (Cat. #: AA433689M; Thermo Fisher Scientific) for 10 min at room temperature, and immediately subjected to immunofluorescence or stored in 1X DPBS at 4°C until use. For each group, 2–3 coverslips were quantified, and 6–10 images were taken from each coverslip using a Nikon Eclipse Ni fluorescent microscope system (Nikon). Images were saved in nd2 or tiff files for further analysis using Visiopharm (Hørsholm) or ImageJ (NIH) software, respectively.

Immunofluorescence for CGRP, NF200, and tubulin $\beta$3 was performed on fixed DRG cells. After blocking with 5% normal donkey serum in 0.03% Triton X-100 in PBS for 1 h at room temperature, the cells were incubated with primary antibodies (anti-CGRP antibody [1:2,000, Cat. #: C8198; MilliporeSigma], anti-NF-200 antibody [1:3,000, Cat. #: CH22104; Neuromics], or anti-tubulin $\beta$3 [TUBB3] antibody [1:1,000, Cat. #: 801201; BioLegend]) overnight at 4°C. The cells were then labeled with the secondary antibodies (CY3-conjugated donkey anti-rabbit IgG [1:600, Cat. #: 711-165-152; Jackson ImmunoResearch], CY5-conjugated donkey anti-chicken IgY [1:400, Cat. #: 703-175-155; Jackson ImmunoResearch], or CY2-conjugated

donkey anti-mouse IgY [1:500, Cat. #: 715-225-150; Jackson ImmunoResearch]) for 2 h at room temperature. After washing five times with 1X DPBS, the cells were mounted with ProLong Gold Antifade Mountant with DAPI. All images were taken using a Nikon Eclipse Ni fluorescent microscope system. For quantification, 6–10 images each coverslip x 2-3 coverslips were randomly selected, and the length of nerve fibers was analyzed, using image analysis software, Nikon Elements V4.13 Basic Research.

### Co-culture between cancer cells and DRG cells

Cancer cells (2,000 cells/60 $\mu$l of growth medium supplemented with 10% FBS) were seeded onto the middle of $\mu$-Slide 2 Well Co-Culture plates (Cat. #: 81806; ibidi). Murine primary DRG sensory nerve cells (1,000 cells/60 $\mu$l of NG medium) were then seeded onto the eight wells surrounding the tumor cells. After 24 h of incubation, 600 $\mu$l of serum-free media was added to allow these two cell types to exchange their secreted molecules. In some cases, co-cultures were treated with either isotype control (100 nM; TEVA Pharmaceuticals) or mouse anti-CGRP monoclonal antibody (100 nM; TEVA Pharmaceuticals). After 72 h of co-culture, cancer cells were stained with 1x crystal violet solution (Cat. #: S25275; Thermo Fisher Scientific [Fisher Science Education]) for 10 min at room temperature, rinsed with tap water five times, and lysed with 100 $\mu$l of 1% SDS solution (Cat. #: 11667289001; MilliporeSigma). The resulting cell lysates were transferred to a clear bottom 96-well plate (Cat. #: 07-200-565; Thermo Fisher Scientific [Corning]), and OD was measured at 570 nm by Epoch Microplate Spectrophotometer (Agilent Technologies). At the termination of the experiments, the resulting DRG cells were also fixed with 4% PFA for 10 min at room temperature, and the nerve growth was confirmed using IF.

### Human samples

Serum samples were obtained from prostate cancer patients with/without bone metastasis from 2016 to 2018 (patients without [n = 11] and with [n = 22] bone metastasis), and ELISAs were used to measure levels of CGRP (Cat. #: CEA876Hu; Cloud-Clone Corp) and substance P (Cat. #: CEA393Hu; Cloud-Clone Corp) in plasma and bone marrow supernatant.

Human prostate adenocarcinoma tissue microarray (TMA) slides were sectioned (3 $\mu$m thickness) at the Jikei University (Tokyo, Japan). Samples were obtained from 93 patients who received prostatectomy in 2006. Greater than 90% of biopsy cores occupied by cancer were included for further analyses.

Autopsy bone samples obtained from prostate and lung cancer patients with/without bone metastases were collected during 2011–2015 and were sectioned at the Jikei University School of Medicine (5 $\mu$m thickness) (prostate cancer patients without [n = 4] and with [n = 4] bone metastasis and lung cancer patients without [n = 2] and with [n = 2] bone metastasis). Before incubating with the primary antibody, slides from patient samples were baked for 90 min at 60°C and treated with DeCal Retrieval solution (HK089-5K; BioGenex) for antigen retrieval for 30 min at room temperature.

After patient samples were blocked with 3% normal goat serum (Cat. #: 005-000-121; Jackson ImmunoResearch) and 0.3% Triton X-100 (Cat. #: X100-500ml; MilliporeSigma) in 0.01 M PBS for 1 h at

room temperature, the TMA slides and bone slides were incubated with anti-CRLR primary antibody (1:50, Cat. #: 84467; Abcam) overnight at 4°C. Slides were then incubated with biotinylated anti-rabbit IgG secondary antibody (SS Rabbit Link, Cat. #: HK336-5R; BioGenex) for 1 h at room temperature. Antibody detection was performed using a VECTASTAIN ABC kit (Cat. #: PK-6100; Vector Laboratories). The CRLR-positive area (DAB intensity) was visualized using a Nikon Eclipse Ni fluorescent microscope. The CRLR-positive area was measured in three different randomly selected fields. The resulting DAB intensities were quantified by ImageJ software (Schindelin et al, 2012) by converting the DAB intensity number to an optical density (OD, OD = log [255 (max intensity)]/mean intensity).

### Animal tissue immunohistochemistry and IF

Murine bone (paraffin): hematoxylin and eosin staining was performed on paraffin-embedded sections of femurs. IF for total p38 (1:200, Cat. #: 9212; Cell Signaling Technology) and phosphorylated p38 (1:200, Cat. #: 4511; Cell Signaling Technology) was performed on paraffin-embedded sections of femurs. Antigen retrieval was conducted using BioGenex DeCal Retrieval Solution (Cat. #: HK089-5K). After blocking with 3% normal donkey serum and 0.3% Triton X-100 in PBS for 1 h at room temperature, the sections were incubated with the first primary antibody, phosphorylated p38, overnight at 4°C. The sections were then labeled with the first secondary antibody CY5-conjugated donkey anti-rabbit IgG (1:600, Cat. #: 711-175-152; Jackson ImmunoResearch) at room temperature for 2 h. Sections were then blocked with 3% normal rabbit serum (Cat. #: 011-000-120; Jackson ImmunoResearch) and 0.3% Triton X-100 in PBS at room temperature for 2 h to saturate open primary antibody binding sites. Sections were blocked again with AffiniPure Fab fragment donkey anti-rabbit IgG (1:500, Cat. #: 711-007-003; Jackson ImmunoResearch) at room temperature for 2 h. Sections were then labeled with the second primary antibody, anti-total p38, overnight at 4°C. Finally, sections were labeled with the second secondary antibody, CY3-conjugated donkey anti-rabbit IgG (1:600), at room temperature for 2 h. Slides were then mounted for imaging. Other bones were stained for TRAP for osteoclasts (Cat#: 387A; Sigma-Aldrich). Slides were mounted using the aqueous mounting media, Aqua-Mount (Thermo Fisher Scientific), and slide edges were set with clear nail polish. Cells that were pinkish/red in color and along the proximal trabecular bone or proximal periosteum were counted as positive for TRAP as quantification for osteoclasts.

Murine bone (frozen): CGRP and NF200 (1:700, Cat. #: 4680; Abcam) staining was performed on frozen sections of the femur. After blocking with 3% normal donkey serum and 0.3% Triton X-100 in PBS for 1 h at room temperature, the frozen sections were incubated with primary antibodies (anti-CGRP and anti-NF200) overnight at 4°C. The sections were then labeled with the secondary antibodies CY3-conjugated donkey anti-rabbit IgG or CY2-conjugated donkey anti-chicken IgY (1:600, Cat. #: 703-225-155; Jackson Immuno-Research) for 2 h at room temperature. Femur sections were mounted in ProLong Gold Antifade Mountant with DAPI (Cat. #: P36935; Thermo Fisher Scientific [Invitrogen]).

Murine spinal cord: IF for GFAP (1:2,000, Cat. #: Z0334; Dako), GFP (1:1,000, Cat. #: A-11122; Thermo Fisher Scientific [Invitrogen]), and CGRP was performed on fixed sections of the spinal cord. After blocking with 3% normal donkey serum and 0.3% Triton X-100 in PBS for 1 h at room temperature, the frozen sections were incubated with primary antibodies (anti-GFAP, anti-GFP, and anti-CGRP) overnight at 4°C. The sections were then labeled with the secondary antibodies CY3-conjugated donkey anti-rabbit IgG or CY2-conjugated donkey anti-chicken IgY for 2 h at room temperature. Spinal cord sections were dehydrated using different gradients of ethanol (70, 90, and 100%, for 2 min each), cleared with xylene (twice for 2 min each), and coverslipped with DPX mounting media (Cat. #: 06522; MilliporeSigma).

Murine DRG: fixed DRG sections were stained for GFP and CGRP. After blocking with 3% normal donkey serum and 0.3% Triton X-100 in PBS for 1 h at room temperature, the frozen sections were incubated with primary antibodies (anti-GFP and anti-CGRP) overnight at 4°C. The sections were then labeled with the secondary antibodies CY3-conjugated donkey anti-rabbit IgG or CY2-conjugated donkey anti-chicken IgY for 2 h at room temperature. DRG sections were mounted in ProLong Gold Antifade Mountant with DAPI.

All images were taken using a Nikon Eclipse Ni fluorescent microscope system (Nikon). Images were then quantified using Visiopharm (Hørsholm), Nikon Elements V4.13 Basic Research, and/or ImageJ software (version 1.51f; National Institutes of Health, Bethesda, MD).

### Bioinformatics

The Cancer Genome Atlas was used to access genomics data available for prostate cancer cohorts (n = 390). Means and standard deviations were presented for continuous characteristics. Medians were also presented in the case that the continuous characteristics are not normally distributed. Counts and percentages were presented for discrete characteristics. The associations between the Gleason score and CALCRL, RAMP1, RAMP2, and RAMP3 were calculated using Spearman's rank correlation coefficients using SAS software (SAS Inc., Cary, NC). The Gleason score was further categorized into three groups (i.e., 6, 7, and 8–10) based on clinically meaningfully cutoff points. Multinomial logistic regression was used to explore the association between CALCRL and the categorized Gleason score. Three models were performed. Model 1 was unadjusted. Model 2 adjusted for RAMP1 and age at diagnosis. Model 3 in addition adjusted for PSA. Odds ratios and their 95% confidence intervals were presented where the Gleason score 6 was treated as the reference group. Recurrence-free survival curves by CALCRL groups (using the median as a cutoff point) were computed using the Kaplan–Meier estimates and compared using log-rank tests. The association between the binary CALCRL (>median versus ≤median) and recurrence-free survival was performed using the Cox proportional hazards model. Three models were performed again with the same parameters. Hazard ratios for the association between binary CALCRL and recurrence-free survival and their 95% confidence intervals were calculated.

The GEO was used to access genomics data available for prostate cancer (GSE6919) (Yu et al, 2004; Chandran et al, 2007) and breast cancer (GSE14017 and GSE14018) (Zhang et al, 2009) cohorts. Log$_2$-transformed gene expression data of CALCRL, RAMP1, RAMP2, and RAMP3 from patient samples were compared between primary

prostate cancer (n = 65) and metastatic prostate cancer (n = 25) obtained from GSE6919; between breast cancer with brain metastasis (n = 15), lung metastasis (n = 4), and bone metastasis (n = 10) obtained from GSE14017; and between breast cancer with brain metastasis (n = 7), lung metastasis (n = 16), liver metastasis (n = 5), and bone metastasis (n = 8) obtained from GSE14017 using GraphPad Prism 9 software (GraphPad Software Inc.).

Benign prostatic hyperplasia (n = 24), primary prostate cancer (n = 33), castration-resistant prostate cancer (CRPC) with soft tissue metastases (n = 129), and CRPC with bone metastases (n = 20) were obtained through the University of Washington Prostate Cancer Donor Autopsy Program (Morrissey et al, 2013). Briefly, tissue was sectioned, and RNA was isolated and amplified as we performed previously (Kumar et al, 2016). Probe labeling was performed using a custom Agilent 44K microarray kit (Cat. #: G4413A) against a reference pool of common prostate cancer cell lines. The CRPC molecular profiling data have been deposited in GEO with the accession number GSE77930 (Kumar et al, 2016).

### ELISA

The levels of CGRP in the serum and bone marrow supernatant collected with protease inhibitors were measured with a custom CGRP ELISA (TEVA Pharmaceuticals). Briefly, mouse anti-CGRP capture antibody (Bertin Bioreagent, Montigny-le-Bretonneux) was coated on a 96-well plate and incubated overnight at 4°C. The coated plate was washed with wash buffer (PBS with 0.05% Tween-20) and then blocked using the Sword Blocker SBL-501 reagent (Sword Diagnostics) for 1 h. After several washes, serum samples and a range of CGRP standards diluted in Sword Diluent SDI-802 were added to the coated plate with a human anti-CGRP detection antibody (Teva Pharmaceuticals) and incubated at room temperature for 2 h. After multiple washes, captured CGRP analytes were complexed to an HRP-conjugated mouse anti-human antibody (Southern Biotech) at room temperature for 1 h. After several washes, Sword detector reagents consisting of a substrate/ peroxidase mixture were used according to the manufacturer's instructions. Resonance Raman signals generated using Sword reagents were measured at an excitation/emission wavelength of 530 nm/730 nm using a BioTek Cytation 5 microplate reader.

### Quantitative Real-Time PCR (qRT-PCR)

Cells or DRG tissues were lysed, and RNA was harvested with RNeasy Mini Kit (Cat. #: 74104; QIAGEN). The RNA concentrations were determined and subsequently normalized between samples before first-strand cDNA synthesis. First-strand cDNA was synthesized using 0.5 $\mu$g of total RNA using Invitrogen SuperScript II Reverse Transcriptase (Cat. #: 18064022; Thermo Fisher Scientific [Invitrogen]). Real-time qPCR (qRT-PCR) was performed using TaqMan Gene Expression Master Mix (Cat. #: 4369016; Applied Biosystems) and TaqMan Gene Expression Assays (Cat. #: 4331182; Applied Biosystems, Assay IDs: Mm99999915_g1 [Gapdh, mouse GAPDH]; Mm00801463_g1 [Calca, mouse CGRP]; Hs02786624_g1 [Gapdh, human GAPDH]; Hs00907738_m1 [CALCRL, human CALCRL]; Hs00195288_m1 [RAMP1, human RAMP1]; Hs00237194_m1 [RAMP2, human RAMP2]; and Hs00389131_m1 [RAMP3, human RAMP3]). qRT-PCR was run for

50 cycles (95°C for 15 s and 60°C for 1 min) after an initial single cycle of 50°C for 2 min and 95°C for 10 min using CFX96 Touch Real-Time PCR Detection System (Bio-Rad). Data are presented as relative gene expression using the delta–delta Ct method, with Gapdh used as the reference gene.

### Western blotting

Cells were lysed with cOmplete Lysis-M reagent (Cat. #: 4719956001; Roche) supplemented with cocktails of protease inhibitor (Cat. #: 11836170001; Roche) and PhosSTOP (Cat. #: 04906837001; Roche). In some cases, cancer cell lines (PC-3, DU145, MDA MB-231, and A549, $1 \times 10^6$ cells/well in a six-well plate [1 ml]) were serum-starved for 5 h and then treated with 10–100 mM CGRP (Cat. #: 015-02; Phoenix Pharmaceuticals) and/or 1–5 $\mu$M SB203580 (Cat. #: S1076; Selleck Chemicals) for 5, 15, 30, and 60 min before cell lysis. In some cases, cells were pretreated with the CGRP receptor antagonist (1 nM, CGRP 8-37, Cat. #: 1181; Tocris Bioscience). After 5 min of boiling at 95°C, protein extracts (10–40 $\mu$g of protein per lane) were loaded, separated on SDS–PAGE (4–20% Tris–glycine gradient gels, Cat. #: XP04202BOX; Thermo Fisher Scientific [Invitrogen]), and transferred to a PVDF membrane (0.2 $\mu$m, Cat. #: ISEQ00010; MilliporeSigma). After electrophoresis, proteins were transferred to a 0.2-$\mu$m PVDF membrane. After blocking with 5% nonfat dry milk (Cat. #: 1706404XTU; Bio-Rad) in TBS/Tween-20 buffer (Cat. #: 28360; Thermo Fisher Scientific) for 1 h, the membrane was incubated with the following primary antibodies at 4°C overnight: anti-CRLR antibody (1:300, Cat. #: 84467; Abcam); anti-RAMP1 antibody (1:500, Cat. #: sc-11379; Santa Cruz Biotechnology); anti-RAMP2 antibody (1:500, Cat. #: ab198276; Abcam); anti-RAMP3 antibody (1:500, Cat. #: ab78017; Abcam), anti-p38 antibody (1:1,000, Cat. #: 9212; Cell Signaling Technology); anti-phospho-p38 (Thr180/Tyr182) antibody (1:1,000, Cat. #: 9211; Cell Signaling Technology); anti-HSP27 antibody (1:1,000, Cat. #: 95357; Cell Signaling Technology); anti-phospho-HSP27 antibody (1:1,000, Cat. #: 9709S; Cell Signaling Technology); or anti-GAPDH antibody (1:1,000, Cat. #: 2118; Cell Signaling Technology). Thereafter, blots were incubated with anti-rabbit IgG HRP-conjugated secondary antibody (1:2,000, Cat. #: 7074S; Cell Signaling Technology) at room temperature for 1 h. Protein expression was detected with Pierce ECL Western Blotting Substrate (Cat. #: 32209; Thermo Fisher Scientific). The densitometry analysis of the Western blot was performed with ImageJ software (National Institutes of Health [NIH], Bethesda, MD).

### Cell proliferation assays

Cells were seeded into a 96-well plate in 100 $\mu$l complete medium at a concentration of $2 \times 10^3$ cells/well. After 24 h of serum starvation, cells were incubated with CGRP (0–1$\mu$M; MilliporeSigma) for 48 h. MTT (3-(4,5-dimethylthiazol-2-yl)-2,5-diphenyltetrazolium bromide, 10 $\mu$l of 5 mg/ml) was added into each well and incubated for 4 h at 37°C, 5% $CO_2$, and 100% humidity, and then, 100 $\mu$l of the solubilization solution was added into each well for overnight. The absorbance of solubilized formazan crystals at 560–650 nm was measured with a microplate spectrophotometer. In some cases, cells were pretreated with the CGRP receptor antagonist (1 nM, CGRP 8-37, Cat. #: 1181; Tocris Bioscience) or p38 inhibitor (5 $\mu$M, SB203580,

Cat. #: S1076; Selleckchem) for 1 h before the CGRP treatment. In other cases, cells were treated with isotype control and anti-CGRP Ab.

Cells were seeded into a 96-well plate in 100 $\mu$l complete medium at a concentration of $2 \times 10^3$ cells/well. After 24 h of serum starvation, cells were incubated with CGRP (0–1 $\mu$M, 90954-53-3; MilliporeSigma) for 72 h. Then, plates were placed in IncuCyte ZOOM System (Essen Bioscience). Phase-contrast pictures were captured every 6 h. Cell confluency (%) and doubling time (hours) of each picture were automatically measured over time, and data were normalized with initial cell confluency. In some cases, cells were pretreated with CGRP 8-37 (dose) for 1 h before the CGRP treatment.

### Intracellular signaling array

Cells were seeded into a six-well plate at $1 \times 10^5$ cells/well. After 24 h of serum starvation, cells were treated with either dH$_2$O or 10 nM CGRP for 0.5, 1, 8, or 24 h. Then, the activation of signaling molecules in the cell lysates (0.3 mg/ml) was determined using PathScan intracellular Signaling Array Kit (Cat. #:7323; Cell Signaling Technology), according to the manufacturer's instruction. The resulting spot intensities were quantified by ImageJ software. The data were normalized by total protein measured using the Pierce BCA protein assay kit (Cat #: 23227; Thermo Fisher Scientific).

### Statistical analyses

Numerical data are expressed as the mean ± SD or SEM. Statistical analysis was performed using GraphPad Prism and SAS statistical program with significance at $P \leq 0.05$. Outcome measures were transformed to satisfy the conditional normality assumption as needed. An unpaired $t$ test or one-way analysis of variance (ANOVA) with Tukey's or Dunnett's post hoc test was used to compare single measurements between groups. For outcome measures (e.g., log-transformed radiance and adjusted guarding time) collected repeatedly over time, mixed-effects models were used to compare mean differences between groups (e.g., DU145 and Sham groups) over time. Group, time, and group-by-time interaction were included in the model. Animals were treated as a random effect. Contrasts were calculated to compare mean differences at each time point.

### Online supplemental material

Fig S1 shows CALCA, RAMP1, RAMP2, and RAMP3 expression in one human prostate cancer cohort (GSE6919) and two human breast cancer cohorts (GSE14017, GSE14018). Fig S2 shows the relative mRNA and protein expression of CALCRL, RAMP1, RAMP2, and RAMP3 in human metastatic cancer cell lines. Fig S3 shows MTT of CGRP treatment on RM-1 cancer cells, depicts representative images of the antibody-based cell pathway array data initially shown in Fig 8A, and validates downstream targets in RM-1. Fig S4 illustrates $\mu$CT scans of femurs from WT mice and CGRP heterozygous (CGRP Control) and homozygous (CGRP KO) mice. In addition, this figure quantifies the trabecular area, cortical area, and femoral neck area in these bones. Fig S5 demonstrates no change in tumor growth between isotype control and anti-CGRP Ab in A549-bearing mice.

## Supplementary Information

## Acknowledgements

We would like to thank Ms. Carol A Aschenbrenner, Ms. Macie D Wilson, Ms. Rebecca M Cain, Ms. Renee Parker, Dr. D Brooke Widner, and Dr. Qingxia Zhao (Wake Forest University School of Medicine); Dr. Shin Egawa and Dr. Hiroyuki Takahashi (Jikei University School of Medicine); and Dr. Daniella Bianchi-Frias (Fred Hutchinson Cancer Center) for the constructive discussion and technical support. The results shown in Fig 3 were in part based upon data generated by TCGA Research Network: https://www.cancer.gov/tcga. This work was directly supported by the National Cancer Institute (R01CA238888, Y Shiozawa; 5 P50CA097186, PS Nelson; R01CA234715, PS Nelson; and R01CA266452, PS Nelson), Department of Defense (W81XWH-17-1-0541, Y Shiozawa; and W81XWH-19-1-0045, Y Shiozawa); METAvivor (METAvivor Research Award, Y Shiozawa); Teva Pharmaceuticals (Y Shiozawa); and the Wake Forest Baptist Comprehensive Cancer Center Internal Pilot Funding (Y Shiozawa). Research reported in this publication was supported by the National Cancer Institute's Cancer Center Support Grant award number P30CA012197 issued to the Wake Forest Baptist Comprehensive Cancer Center. The authors wish to acknowledge the support of the Wake Forest Baptist Comprehensive Cancer Center Cell Engineering, Flow Cytometry, and Biostatistics Shared Resources, supported by the National Cancer Institute's Cancer Center Support Grant award number P30CA012197. The content is solely the responsibility of the authors and does not necessarily represent the official views of the National Cancer Institute and Department of Defense.

### Author Contributions

SH Park: conceptualization, data curation, formal analysis, and writing—original draft.

S Tsuzuki: conceptualization, data curation, formal analysis, and writing—original draft.

KF Contino: conceptualization, data curation, formal analysis, investigation, methodology, and writing—original draft, review, and editing.

J Ollodart: conceptualization, data curation, formal analysis, investigation, methodology, and writing—original draft, review, and editing.

MR Eber: data curation and formal analysis.

Y Yu: data curation and formal analysis.

LR Steele: data curation and formal analysis.

H Inaba: resources.

Y Kamata: resources, data curation, and formal analysis.

T Kimura: resources.

I Coleman: resources, data curation, and formal analysis.

PS Nelson: resources, data curation, and formal analysis.

E Muñoz-Islas: data curation and formal analysis.

JM Jiménez-Andrade: data curation and formal analysis.

TJ Martin: resources.

KD Mackenzie: resources, data curation, formal analysis, investigation, methodology, and writing—review and editing.

JR Stratton: resources, data curation, formal analysis, investigation, methodology, and writing—review and editing.

F-C Hsu: formal analysis.

CM Peters: data curation, formal analysis, investigation, and methodology.

Y Shiozawa: conceptualization, resources, data curation, formal analysis, supervision, funding acquisition, investigation, visualization, methodology, and writing—original draft, review, and editing.

## Conflict of Interest Statement

Part of the study was funded by Teva Pharmaceuticals (Yusuke Shiozawa). KD Mackenzie and JR Stratton were employed by Teva Pharmaceuticals. No conflict of interest exists for the remaining authors.

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
