## [Reviewer comments · Life Science Alliance]

Life Science Alliance

Crosstalk between bone metastatic cancer cells and sensory nerves in bone metastatic progression

Sun H Park, Shunsuke Tsuzuki, Kelly F Contino, Jenna Ollodart, Matthew R Eber, Yang Yu, Laiton R Steele, Hiroyuki Inaba, Yuko Kamata, Takahiro Kimura, Ilsa Coleman, Peter S Nelson, Enriqueta Muñoz-Islas, Juan Miguel Jiménez-Andrade, Thomas J Martin, Kimberly D Mackenzie, Jennifer R Stratton, Fang-Chi Hsu, Christopher M Peters and Yusuke Shiozawa

DOI: <https://doi.org/10.26508/lsa.202302041>

Corresponding author(s): Dr. Yusuke Shiozawa (Wake Forest University School Of Medicine)

Review Timeline:

Submission Date:	2023-03-15
Editorial Decision:	2023-04-26
Revision Received:	2024-06-24
Editorial Decision:	2024-08-01
Revision Received:	2024-08-29
Editorial Decision:	2024-09-04
Revision Received:	2024-09-05
Accepted:	2024-09-06

Transaction Report:

April 26, 2023

Re: Life Science Alliance manuscript #LSA-2023-02041-T

Dr. Yusuke Shiozawa
Wake Forset
Winston-Salem

Dear Dr. Shiozawa,

Thank you for submitting your manuscript entitled "Crosstalk between bone metastatic cancer cells and sensory nerves in bone metastatic progression" to Life Science Alliance. The manuscript was assessed by expert reviewers, whose comments are appended to this letter. We invite you to submit a revised manuscript addressing the Reviewer comments.

Thank you for this interesting contribution to Life Science Alliance. We are looking forward to receiving your revised manuscript.

Sincerely,

B. MANUSCRIPT ORGANIZATION AND FORMATTING:

Reviewer #1 (Comments to the Authors (Required)):

This paper is focused on emphasizing the role of CGRP in bone metastasis, a molecule important for vasodilation predominantly secreted by neuronal cells. The authors aim to show that CGRP promotes proliferation of bone metastatic cell lines in comparison to non-metastatic cancer cells, contributes to bone pain and bone remodeling (shown in knockout mice) and suppresses growth of prostate cancer in bone. Several phosphokinases are altered after incubation with CGRP, including p38 and Hsp27, which the authors deem as a mechanism of CGRP-induced proliferation in cancer cells. Although the findings are interesting, the main conclusions are largely descriptive, lacks mechanistic insight, and is somewhat unfocused (there's a substantial mix of testing of different cell lines but little consistency in which particular cell lines are tested per experiment and justification for the inconsistency). This manuscript needs major revision and additional experiments to support the major conclusions and advance the field in this area. It is not yet ready for publication until the concerns are addressed.

1) Although the focus seems to be generally on bone metastasis, there's no comment on differences between the different cancer cell lines that contribute significantly to heterogeneity in the bone phenotype. Further, the bulk of the patient and in vivo data is prostate cancer focused. Prostate cancer bone mets are mostly osteogenic; however, PC3 and DU145 promote osteolysis as well as MDA-MB231. RM1 cells typically induce mixed osteogenic and osteolytic lesions in vivo. The authors should discuss these differences in detail with regards to how the cancer-associated bone phenotype might be impacted by heightened CGRP levels. Further, please include analysis of other CGRP in other cancer patient datasets (in addition to the already included prostate cancer) to provide insight into the pan-cancer relevance of CGRP.

2) A major finding of the paper is that CGRP increases cancer cell proliferation and this seems to be through p38 and HSP27 phosphorylation. However, additional experiments are needed to make this conclusion. The authors showed that inhibition of p38 suppressed cell proliferation induced by CGRP but this still could be an indirect trend. If the CGRP receptor, CRLR, is inhibited (via drug and/or siRNA), is there reduced p38 and Hsp27? Please show this experiment in support of your conclusion. Additionally, there was literally no change in p38 expression in vivo (Figure 10) suggesting that isn't the mode of activation for the antibody. Were RM1 cells interrogated for CGRP-altered targets or only the human cells? The array hits need to be validated in the RM1 cells if that's the cell line used for the in vivo model

3) A major finding is that inhibition of CGRP using antibodies suppresses prostate cancer growth in bone. However, there was no impact (stated) on tumor growth in the CGRP knockout mice. However, it does look like there was a reduction in growth based on Figure 9F. It's unclear why the authors didn't mention this difference in growth or why there's no explanation of differences in host CGRP and antibody treatment. A previous paper showed that knockdown of CGRP in PC3-2M cells (PMID: 33086087) reduces prostate cancer growth in bone which reduces the novelty of this study. However, it is possible that host vs. tumor CGRP should be explored further for translation of these findings. The authors should discuss this in detail. Also, the main scope of the paper seems bone metastasis focused so it's unclear why they chose to only report on prostate cancer patients and also a prostate cancer in vivo model. This paper would be strengthened and be more novel if there were other bone metastasis patient data (from multiple cancer types) and an additional bone metastasis model of another cancer performed with the CGRP antibodies.

4) The authors mention that there were no bone changes in the in vivo antibody study but this was by microCT analysis. What about histomorphologic changes seen in Trichrome or TRAP staining. Please show this supporting data for the mouse knockout model and/or the antibody study since the knockout mouse showed a change in bone development. Mechanistic insight into the reason for this phenotype is needed.

5) There's a lot of inconsistency in the cell lines used. The paper seems to be focused mostly on bone metastatic prostate cancer, so then why were there so many other cell lines used and PC3 was only used in some experiments and then replaced with DU145 and LNCaP in others. Why? Why were conclusions made from DU145 in some experiments and PC3 in others? DU145 is technically from a brain metastasis whereas PC3 is from a bone met and should be used in all experiments. Please update any necessary experiments to include PC3 or justify the reason for not using that cell line in some experiments.

Reviewer #2 (Comments to the Authors (Required)):

In this paper, Park et al. studied the crosstalk between bone metastatic cells and sensory nerves, which is less appreciated in early studies. They demonstrated that by releasing CGRP, sensory nerves may activate CRLP expressed on cancer cells and promote the bone metastatic progression through p38/HSP27. Hence, CGRP/CRLR axis represents a potential therapeutic target for bone metastasis treatment. The topic of this work is intriguing; the effort and labor being devoted to this work is impressive. However, several major caveats in regards to the model utilization, research strategy and science rationale were identified, which need to be thoroughly addressed before being considered for publication in Life Science Alliance.

Fig. 1:

The authors are recommended to include an extra group like orthotopic tumor, or at least subcutaneous tumor, in addition to Sham mice to demonstrate that the presented phenotypes are specifically raised by bone metastasis other than the general outcome of tumor growth in any organs.

Fig. 2:

The notion of "cancer cell lines known to metastasize to the bone" is misleading. A549 is derived from lung cancer and DU145 is collected from brain metastasis. It is conceivable that they should exhibit more growth trait in lung and brain instead of in bone. Indeed, the potential to grow in bone is not necessarily equal to the phenotype of bone metastatic growth. Reasoning that metastasis involves a selection process, the ultimate bone colonization is probably established by a small proportion of subclones, which could not represent the majority of parental cell lines. To address this caveat, the authors are recommended to use 1, cell lines developed from bone metastasis such as C4-2B in addition to PC3; 2, Bone derivatives developed by other researchers in this field such as MDA-231-SCP28 or MDA-231-1833 and compare with other derivatives with different metastatic tropism.

Fig. 3-4:

The results about RAMP1-3 are distractive and redundant. The repeated rejection of their involvement does not help to strengthen the paper's central hypothesis.

With the demonstration of Fig. 4M, Fig. 4A becomes redundant and less relevant.

Fig. 5:

Fig. 4M shows the CALCRL expression is almost identical in benign prostate and primary tumor. By contrast, Fig. 5B suggested CRLR density in prostate tumor is higher than the adjacent (benign) tissues. These results are somewhat conflicting. Given that CRLR is proposed to express on cancer cells, it is fully expected that metastases-free bone marrow should not show any positive CRLR staining. Thus, the rationale to present data 5E-G is confusing, which does not help to strengthen the paper's central hypothesis.

Fig. 6-8:

Again, the grouping by the notion of "known to metastasize to bone" is problematic. Similar concern has been expressed under Fig. 2. Additionally, considering the inherited difference of these cell lines with various background of different cancer types, it is difficult to determine whether the observed changes are truly pertinent to CRLR signaling. A more common and reasonable approach (for all the experiments in Fig. 6-8) is genetic depletion of CRLR in selected cell lines in comparison to the Sham control.

The same strategy (CRLP-depletion) should also be applied to in vivo experiment to determine whether this receptor is truly contributing to bone metastatic progression. This is a critical experiment that was missing.

In addition to the cell models, the rationale underlying the proliferation assays in Fig. 6-8 is also questionable. Proliferation in 2D culture does not necessarily represent the metastatic phenotype in bone. The authors are recommended to use either more relevant cell models or create experimental settings mimicking bone environments.

Fig. 9:

The label "CGRP Homozygous" is misleading, which is supposed to be "KO homozygous" or "CGRP knockout".

The whole Fig. 9, without the involvement of tumor inoculation, is likely supportive data describing the phenotype of GEMM, which is not directly relevant to the major topic. These data are suggested to be moved to supplementary.

Fig. 10:

Though the reviewer appreciates the authors' honesty, the results of CGRP KO and antibody neutralizing are disappointed as determinant experiments. It is also hard to explain why the global knockout of CGRP failed to achieve a comparable effect to monoclonal antibody treatment.

Reviewer #3 (Comments to the Authors (Required)):

The paper titled "Crosstalk between bone metastatic cancer cells and sensory nerves in bone metastatic progression" investigates the role of cancer/CGRP-expressing sensory nerve in bone metastatic progression. The authors have successfully quantified CGRP and CRLR expression in mice and patients with bone metastasis and evaluated the efficacy of blocking CGRP to reduce cancer cell proliferation in vitro and bone metastatic progression in vivo. Overall, the topic is important and the study is rigor. The paper highlights the potential therapeutic targets of CGRP and its receptor CRLR axis. Following is a list items the author may consider:

1. Including merged images would be advantageous. The images of bone marrow (contralateral vs. ipsilateral, CGRP) seem to show a difference in CGRP positive sensory nerves, contradicting the authors' claim that there is no difference. Additionally, it is unclear why the images for the same tissue from both sides of the body look so different.
2. Figure 5(E-G): The quantification includes both bone marrow and cancer cells, but the IHC images do not clearly label the corresponding areas, causing confusion. It is difficult to discern from the images the tissue where CRLR is overexpressed. Furthermore, the 20X images for Bone mets(-) and Bone mets(+) do not appear to be on the same scale.

3. Figures 6-7: The authors conclude that the proliferation rates of lower metastatic potential cell lines LNCaP and MCF-7 are not affected by CGRP but do not explain if this is due to a lack of CRLR receptor in these two cell lines. Including the mRNA levels of CRLR in these two cell lines would strengthen the conclusion in Figure 8. Moreover, the assertion that substance P does not alter cancer cell proliferation is insufficient to conclude that CGRP is solely responsible for this effect, as other proteins, such as cytokines, might be involved in CIBP and secreted by nerves to potentially affect cancer cell proliferation. Additionally, Figure 7C suggests that some cell lines might indeed be affected by substance P.

4. The authors may consider including a dose-response curve to calculate the EC50, as the concentration of CGRP seems to impact cancer cell lines differently. When investigating the mechanism of CGRP-mediated cancer cell proliferation, it is important to clarify why 10 nM is used and whether the concentration could affect the results of mechanism studies.

5. Figure 10: Including CGRP concentration in bone marrow/tumor after antibody administration would provide more substantial support for the conclusion that blocking CGRP from sensory nerves can reduce bone metastatic progression. Figures H, I, and J need further clarification and additional experiments to strengthen the claims made. These additional experiments could include: a) verifying that the CGRP antibody itself does not affect cancer cell proliferation, b) measuring CGRP concentration in the medium with/without the antibody, and c) demonstrating that the presence of cancer cells increases the neurite outgrowth length of DRGs, which would be consistent with previous findings in the paper (medium from cancer cell lines increase DRG neurite length).

6. On page 25, the authors mention Raman, but mentioned that the signal was measured using fluorescence intensity. Raman is not measured using fluorescence intensity. Please check,

7. at a Discussion: The authors might consider discussing the CRLR/p38/HSP27 pathway as alternative therapeutic targets for the anti-CGRP antibody, especially considering the inhibition of CRLR, as many results of the paper indicate high levels of CRLR in tumors are associated with bone metastasis.

Overall, the paper offers important findings and contributes to the field. By addressing the suggestions mentioned above, the authors can further enhance the clarity and impact of their research.

Reviewer's comments:**Reviewer #1:****Comment 1:**

This paper is focused on emphasizing the role of CGRP in bone metastasis, a molecule important for vasodilation predominantly secreted by neuronal cells. The authors aim to show that CGRP promotes proliferation of bone metastatic cell lines in comparison to non-metastatic cancer cells, contributes to bone pain and bone remodeling (shown in knockout mice) and suppresses growth of prostate cancer in bone. Several phosphokinases are altered after incubation with CGRP, including p38 and Hsp27, which the authors deem as a mechanism of CGRP-induced proliferation in cancer cells. Although the findings are interesting, the main conclusions are largely descriptive, lacks mechanistic insight, and is somewhat unfocused (there's a substantial mix of testing of different cell lines but little consistency in which particular cell lines are tested per experiment and justification for the inconsistency). This manuscript needs major revision and additional experiments to support the major conclusions and advance the field in this area. It is not yet ready for publication until the concerns are addressed.

Response:

We thank the reviewer all the efforts to provide insightful comments. We believe that the quality of our manuscript has improved extensively by addressing the reviewer's comments.

Comment 2:

Although the focus seems to be generally on bone metastasis, there's no comment on differences between the different cancer cell lines that contribute significantly to heterogeneity in the bone phenotype. Further, the bulk of the patient and in vivo data is prostate cancer focused. Prostate cancer bone mets are mostly osteogenic; however, PC3 and DU145 promote osteolysis as well as MDA-MB231. RM1 cells typically induce mixed osteogenic and osteolytic lesions in vivo. The authors should discuss these differences in detail with regards to how the cancer-associated bone phenotype might be impacted by heightened CGRP levels.

Response:

We now discussed on the point raised by the reviewer.

Comment 3:

Further, please include analysis of other CGRP in other cancer patient datasets (in addition to the already included prostate cancer) to provide insight into the pan-cancer relevance of CGRP.

Response:

To look into the CGPR levels in patients' serum of other cancer type, such as breast cancer, with/without bone metastasis, we sought to obtain IRB and then collect their serum. However, it is very difficult to complete this task in a such short time. Further, we performed literature search. But we could not find any literatures investigating serum CGRP levels in bone metastatic cancer patients vs. non-bone metastatic cancer patients, other than prostate cancer patients (the manuscripts were already cited in our paper. However, to see if there are any differences in the CGRP levels between primary tumor and metastatic tumor, we included two breast cancer patient

cohorts GSE14017 and GSE14018 and one prostate cancer patient cohort GSE6919 for CGRP analysis (new Supplemental Figure 1). We looked for additional lung and melanoma patient cohorts but ultimately could not find a useful cohort of metastasis. Hope the reviewer understands our honest attempts.

Comment 4:

A major finding of the paper is that CGRP increases cancer cell proliferation and this seems to be through p38 and HSP27 phosphorylation. However, additional experiments are needed to make this conclusion. The authors showed that inhibition of p38 suppressed cell proliferation induced by CGRP but this still could be an indirect trend. If the CGRP receptor, CRLR, is inhibited (via drug and/or siRNA), is there reduced p38 and Hsp27? Please show this experiment in support of your conclusion.

Response:

We performed the experiment suggested by the reviewer, shown in new Figure 9A.

Comment 5:

Additionally, there was literally no change in p38 expression in vivo (Figure 10) suggesting that isn't the mode of activation for the antibody. Were RM1 cells interrogated for CGRP-altered targets or only the human cells? The array hits need to be validated in the RM1 cells if that's the cell line used for the in vivo model.

Response:

We looked at levels of p38, p-p38, HSP27, and p-HSP7 in RM-1 (new Supplemental Fig 3B). We saw no effects of CGRP on p-HSP27 levels in this cell line. We discussed this discrepancy in the revised manuscript.

Comment 6:

A major finding is that inhibition of CGRP using antibodies suppresses prostate cancer growth in bone. However, there was no impact (stated) on tumor growth in the CGRP knockout mice. However, it does look like there was a reduction in growth based on Figure 9F. It's unclear why the authors didn't mention this difference in growth or why there's no explanation of differences in host CGRP and antibody treatment. A previous paper showed that knockdown of CGRP in PC3-2M cells (PMID: 33086087) reduces prostate cancer growth in bone which reduces the novelty of this study. However, it is possible that host vs. tumor CGRP should be explored further for translation of these findings. The authors should discuss this in detail.

Response:

We confirmed with biostatistician that there was no significant reduction in tumor growth in Figure 9F. We discussed on the differences between host CGRP vs tumor CGRP. Please also see above response regarding there is no differences in tumor CGRP levels between primary tumor vs, metastatic tumor (new Supplemental Figure 1).

Comment 7:

Also, the main scope of the paper seems bone metastasis focused so it's unclear why they chose to only report on prostate cancer patients and also a prostate cancer in vivo model. This paper would be strengthened and be more novel if there were other bone metastasis patient data (from multiple cancer types) and an additional bone metastasis model of another cancer performed with the CGRP antibodies.

Response:

Unfortunately, we could not find other bone metastasis patient cohort (lung, breast, melanoma, etc.) other than patient data shown in this paper (new Figure 5). However, we performed another animal study with the anti-CGRP antibody using a human lung cancer cell line (new Supplement Figure 5). Hope the reviewer understands our honest attempts.

Comment 8:

The authors mention that there were no bone changes in the in vivo antibody study but this was by microCT analysis. What about histomorphologic changes seen in Trichrome or TRAP staining. Please show this supporting data for the mouse knockout model and/or the antibody study since the knockout mouse showed a change in bone development. Mechanistic insight into the reason for this phenotype is needed.

Response:

We performed TRAP staining on bone marrow samples from mice treated with anti-CGRP antibody (new Figure 11) and discussed on this phenomenon.

Comment 9:

There's a lot of inconsistency in the cell lines used. The paper seems to be focused mostly on bone metastatic prostate cancer, so then why were there so many other cell lines used and PC3 was only used in some experiments and then replaced with DU145 and LNCaP in others. Why? Why were conclusions made from DU145 in some experiments and PC3 in others? DU145 is technically from a brain metastasis whereas PC3 is from a bone met and should be used in all experiments. Please update any necessary experiments to include PC3 or justify the reason for not using that cell line in some experiments.

Response:

We made an effort to include PC3 in most of the experiments. We chose DU145 as a model to study cancer-induced bone pain in Figure 1 and 2, since DU145 showed more pain behavior than PC3 (preliminary study). Further, DU145 induced more nerve sprouting than PC3 (new Figure 2D). We included this information in our revised manuscript.

Reviewer #2:

Comment 1:

In this paper, Park et al. studied the crosstalk between bone metastatic cells and sensory nerves, which is less appreciated in early studies. They demonstrated that by releasing CGRP, sensory nerves may activate CRLP expressed on cancer cells and promote the bone metastatic progression through p38/HSP27. Hence, CGRP/CRLR axis represents a potential therapeutic target for bone metastasis treatment. The topic of this work is intriguing; the effort and labor being devoted to this work is impressive. However, several major caveats in regards to the model utilization, research strategy and science rationale were identified, which need to be thoroughly addressed before being considered for publication in Life Science Alliance.

Response:

We thank the reviewer all the efforts to provide insightful comments. We believe that the quality of our manuscript has improved extensively by addressing the reviewer's comments.

Comment 2:

The authors are recommended to include an extra group like orthotopic tumor, or at least subcutaneous tumor, in addition to Sham mice to demonstrate that the presented phenotypes are specifically raised by bone metastasis other than the general outcome of tumor growth in any organs.

Response:

We performed the experiment suggested by the reviewer (new Figure 1 H&I).

Comment 3:

The notion of "cancer cell lines known to metastasize to the bone" is misleading. A549 is derived from lung cancer and DU145 is collected from brain metastasis. It is conceivable that they should exhibit more growth trait in lung and brain instead of in bone. Indeed, the potential to grow in bone is not necessarily equal to the phenotype of bone metastatic growth. Reasoning that metastasis involves a selection process, the ultimate bone colonization is probably established by a small proportion of subclones, which could not represent the majority of parental cell lines. To address this caveat, the authors are recommended to use 1, cell lines developed from bone metastasis such as C4-2B in addition to PC3; 2, Bone derivatives developed by other researchers in this field such as MDA-231-SCP28 or MDA-231-1833 and compare with other derivatives with different metastatic tropism.

Response:

Per reviewer's suggestion, we changed "cancer cell lines known to metastasize to the bone" to "capable of metastasizing to the bone". Further, we have tried to develop C4-2B mouse model, however, cells are difficult to colonize and grow in vivo. Hope the reviewer understands our honest attempts.

Comment 4:

The results about RAMP1-3 are distractive and redundant. The repeated rejection of their involvement does not help to strengthen the paper's central hypothesis. With the demonstration of Fig. 4M, Fig. 4A becomes redundant and less relevant.

Response:

Per reviewer's suggestion, we moved the data regarding RAMP1-3 patient data to new Supplemental Figure 1.

Comment 5:

Fig. 4M shows the CALCRL expression is almost identical in benign prostate and primary tumor. By contrast, Fig. 5B suggested CRLR density in prostate tumor is higher than the adjacent (benign) tissues. These results are somewhat conflicting. Given that CRLR is proposed to express on cancer cells, it is fully expected that metastases-free bone marrow should not show any positive CRLR staining. Thus, the rationale to present data 5E-G is confusing, which does not help to strengthen the paper's central hypothesis.

Response:

Figure 4M (now new Figure 5D) is comparing the RNA expression of CRLR whereas Fig 5B (now new Figure 6B) is showing the quantification of IHC staining for the CRLR protein. RNA expression and protein levels may not always directly correspond. Further, the samples are from different patient populations from different cohorts, which could also have an impact on CRLR levels.

CRLR is expressed by many cell types, including normal bone resident cells such as bone marrow cells (osteocytes), which is why metastases-free bone marrow still has some low levels of CRLR staining; however, this expression is substantially higher in bone metastatic cancer cells.

We made these two points clear in our revised manuscript.

Comment 6:

Again, the grouping by the notion of "known to metastasize to bone" is problematic. Similar concern has been expressed under Fig. 2.

Response:

We rephrased problematic phrase to "which are capable of metastasizing to bone" (see also response above).

Comment 7:

Additionally, considering the inherited difference of these cell lines with various background of different cancer types, it is difficult to determine whether the observed changes are truly pertinent to CRLR signaling. A more common and reasonable approach (for all the experiments in Fig. 6-8) is genetic depletion of CRLR in selected cell lines in comparison to the Sham control. The same strategy (CRLP-depletion) should also be applied to in vivo experiment to determine whether this receptor is truly contributing to bone metastatic progression. This is a critical experiment that was missing.

Response:

We performed animal study with CGRP8-37 (CGRP receptor antagonist) to inhibit binding between CGRP and CRLR, instead of CRLR knockdown and discussion on targeting CGRP pathway as a potential therapy.

Comment 8:

In addition to the cell models, the rationale underlying the proliferation assays in Fig. 6-8 is also questionable. Proliferation in 2D culture does not necessarily represent the metastatic phenotype in bone. The authors are recommended to use either more relevant cell models or create experimental settings mimicking bone environments.

Response:

Bone is a dynamic microenvironment, unfortunately, there are no *in vitro* models that can recapitulate this type of setting, and therefore relevant models do not exist and 2D culture is the best option available. However, we discuss this notion in our revised manuscript.

Comment 9:

The label "CGRP Homozygous" is misleading, which is supposed to be "KO homozygous" or "CGRP knockout". The whole Fig.9, without the involvement of tumor inoculation, is likely supportive data describing the phenotype of GEMM, which is not directly relevant to the major topic. These data are suggested to be moved to supplementary.

Response:

Per reviewer's suggestion we changed the labels of these mice. We did perform tumor inoculation in GEMM and presented data of bone remodeling, guarding, and tumor growth in new Figure 10D-F. We elect to leave new Figure 10A-C as original figures, despite without the involvement of tumor inoculation, since we believe that this information is crucial to indicate the phenotype of these mice.

Comment 10:

Though the reviewer appreciates the authors' honesty, the results of CGRP KO and antibody neutralizing are disappointed as determinant experiments. It is also hard to explain why the global knockout of CGRP failed to achieve a comparable effect to monoclonal antibody treatment.

Response:

We discuss this notion in our revised manuscript.

Reviewer #3:

Comment 1:

The paper titled "Crosstalk between bone metastatic cancer cells and sensory nerves in bone metastatic progression" investigates the role of cancer/CGRP-expressing sensory nerve in bone metastatic progression. The authors have successfully quantified CGRP and CRLR expression in mice and patients with bone metastasis and evaluated the efficacy of blocking CGRP to reduce cancer cell proliferation in vitro and bone metastatic progression in vivo. Overall, the topic is important and the study is rigor. The paper highlights the potential therapeutic targets of CGRP and its receptor CRLR axis.

Response:

We thank the reviewer all the efforts to provide insightful comments. We believe that the quality of our manuscript has improved extensively by addressing the reviewer's comments.

Comment 2:

Including merged images would be advantageous. The images of bone marrow (contralateral vs. ipsilateral, CGRP) seem to show a difference in CGRP positive sensory nerves, contradicting the authors' claim that there is no difference. Additionally, it is unclear why the images for the same tissue from both sides of the body look so different.

Response:

We included merged images of the bone marrow with CGRP and NF200 staining and changed the representative image to fit with the quantification data (new Figure 2A).

Comment 3:

Figure 5(E-G): The quantification includes both bone marrow and cancer cells, but the IHC images do not clearly label the corresponding areas, causing confusion. It is difficult to discern from the images the tissue where CRLR is overexpressed. Furthermore, the 20X images for Bone mets(-) and Bone mets(+) do not appear to be on the same scale.

Response:

Per reviewer's suggestion, we included annotations of the bone marrow and cancer cells in images. We also confirmed that the scales are the same for the 20X images (new Figure 6).

Comment 4:

Figures 6-7: The authors conclude that the proliferation rates of lower metastatic potential cell lines LNCaP and MCF-7 are not affected by CGRP but do not explain if this is due to a lack of CRLR receptor in these two cell lines. Including the mRNA levels of CRLR in these two cell lines would strengthen the conclusion in Figure 8.

Response:

We thank the reviewer for this suggestion. Per the reviewer's suggestion, we included LNCaP and MCF-7 mRNA expression of CRLR and RAMP1-3 (new Supplemental Figure 2).

Comment 5:

Moreover, the assertion that substance P does not alter cancer cell proliferation is insufficient to conclude that CGRP is solely responsible for this effect, as other proteins, such as cytokines, might be involved in CIBP and secreted by nerves to potentially affect cancer cell proliferation. Additionally, Figure 7C suggests that some cell lines might indeed be affected by substance P.

Response:

We removed “solely” in order to convey that CGRP is one pathway that is responsible for cancer cell proliferation. Additionally, while SP does have a significant effect in two cell lines, this a decrease in proliferation of cancer cells.

Comment 6:

The authors may consider including a dose-response curve to calculate the EC50, as the concentration of CGRP seems to impact cancer cell lines differently. When investigating the mechanism of CGRP-mediated cancer cell proliferation, it is important to clarify why 10 nM is used and whether the concentration could affect the results of mechanism studies.

Response:

We now provided data required by the reviewer (new supplemental Figure 3C).

Comment 7:

Including CGRP concentration in bone marrow/tumor after antibody administration would provide more substantial support for the conclusion that blocking CGRP from sensory nerves can reduce bone metastatic progression. Figures H, I, and J need further clarification and additional experiments to strengthen the claims made. These additional experiments could include: a) verifying that the CGRP antibody itself does not affect cancer cell proliferation, b) measuring CGRP concentration in the medium with/without the antibody, and c) demonstrating that the presence of cancer cells increases the neurite outgrowth length of DRGs, which would be consistent with previous findings in the paper (medium from cancer cell lines increase DRG neurite length).

Response:

We included a MTT assay for anti-CGRP antibody and isotype verifying that the antibody itself does not enhance proliferation (new Figure 12E&F). Unfortunately, measuring CGRP concentration in the medium cannot be done due to interference of anti-CGRP antibody used for treatment with anti-CGRP antibody used for ELISA. Neurite outgrowth assay presented in Figure 10G-J (now new Figure 11 A-C) were performed under the presence of cancer.

Comment 8:

On page 25, the authors mention Raman, but mentioned that the signal was measured using fluorescence intensity. Raman is not measured using fluorescence intensity. Please check,

Response:

We fixed this.

Comment 9:

At a Discussion: The authors might consider discussing the CRLR/p38/HSP27 pathway as alternative therapeutic targets for the anti-CGRP antibody, especially considering the inhibition of CRLR, as many results of the paper indicate high levels of CRLR in tumors are associated with bone metastasis. Overall, the paper offers important findings and contributes to the field. By addressing the suggestions mentioned above, the authors can further enhance the clarity and impact of their research.

Response:

We discuss this notion in our revised manuscript.

August 1, 2024

Re: Life Science Alliance manuscript #LSA-2023-02041-TR

Dr. Yusuke Shiozawa
Wake Forset University School Of Medicine
Medical Center Blvd
Winston-Salem 27157

Dear Dr. Shiozawa,

Thank you for submitting your revised manuscript entitled "Crosstalk between bone metastatic cancer cells and sensory nerves in bone metastatic progression" to Life Science Alliance. The manuscript has been seen by the original reviewers whose comments are appended below. While the reviewers continue to be overall positive about the work in terms of its suitability for Life Science Alliance, some important issues remain.

Our general policy is that papers are considered through only one revision cycle; however, given that the suggested changes are relatively minor, we are open to one additional short round of revision. Please note that I will expect to make a final decision without additional reviewer input upon re-submission.

Please submit the final revision within one month, along with a letter that includes a point by point response to the remaining reviewer comments.

To upload the revised version of your manuscript, please log in to your account: <https://lsa.msubmit.net/cgi-bin/main.plex>
You will be guided to complete the submission of your revised manuscript and to fill in all necessary information.

B. MANUSCRIPT ORGANIZATION AND FORMATTING:

Sincerely,

Reviewer #2 (Comments to the Authors (Required)):

I appreciate the authors' efforts to address the reviewer's comments. However, several issues remain in the study: Firstly, many of the comments were not fully addressed due to challenges with the models and methodologies used. While the

reviewer acknowledges these difficulties, some issues are addressable. For instance, the authors argue that "there are no in vitro models that can recapitulate this type of setting, and therefore relevant models do not exist, making 2D culture the best available option." This is incorrect. Even if advanced engineering biofabricated models are not accessible, there are several alternative bone metastasis in vivo models available, such as 2D/3D co-culture models (PMID: 24145351, PMID: 32824479). Secondly, the rigor of the data needs substantial improvement. For example, the Western blot data using the CGRP antibody (shown in the left 4 lanes) should be consistent between Figures 8B and 9A. However, the patterns in these figures are discrepant. Additionally, the inter-group differences in most of the in vivo experiments (Figures 9 through 11) are marginal and lack statistical significance, making it difficult to support the proposed hypothesis. Overall, while the authors have made a considerable effort to revise the manuscript based on the reviewers' feedback, the conclusions and interpretations presented extend beyond the data provided and require adjustment.

Reviewer #3 (Comments to the Authors (Required)):

The authors answered all questions. The manuscript can now be accepted

Reviewer's comments:**Reviewer #2:****Comment 1:**

I appreciate the authors' efforts to address the reviewer's comments. However, several issues remain in the study: Firstly, many of the comments were not fully addressed due to challenges with the models and methodologies used. While the reviewer acknowledges these difficulties, some issues are addressable. For instance, the authors argue that "there are no in vitro models that can recapitulate this type of setting, and therefore relevant models do not exist, making 2D culture the best available option." This is incorrect. Even if advanced engineering biofabricated models are not accessible, there are several alternative bone metastasis in vivo models available, such as 2D/3D co-culture models (PMID: 24145351, PMID: 32824479).

Response:

We thank the reviewer for meaningful comments. We appreciate feedback regarding bone metastasis 2D/3D co-culture models. While creating such models are beyond the scope of the present study, we have added discussion about 3D co-culture models as screening techniques for bone metastasis treatments and for use in our future studies. We have also cited the manuscripts that reviewer suggested.

Comment 2:

Secondly, the rigor of the data needs substantial improvement. For example, the Western blot data using the CGRP antibody (shown in the left 4 lanes) should be consistent between Figures 8B and 9A. However, the patterns in these figures are discrepant.

Response:

We thank the reviewer for this valuable feedback and recognize some minor changes in activation times in Figure 8B and Figure 9A. We hypothesize that this minor variation is due to the heterogeneity of tumor cells, even within the same cell line and the relatively small differences in activation times used in this study (e.g. 15min vs. 30min). Moreover, when performing Western blots, we made all possible efforts to keep experimental parameters consistent (cell density, low cell passage number, treatment times, etc.); however, as these experiments were performed years apart, some parameters such as imaging equipment changes, CGRP lot number, etc. may have also contributed to minor changes in activation. We included further discussion in the paper regarding these points.

Comment 3:

Additionally, the inter-group differences in most of the in vivo experiments (Figures 9 through 11) are marginal and lack statistical significance, making it difficult to support the proposed

hypothesis. Overall, while the authors have made a considerable effort to revise the manuscript based on the reviewers' feedback, the conclusions and interpretations presented extend beyond the data provided and require adjustment.

Response:

We have added discussion regarding the marginal differences in Figure 9-11 and reasoning for why there was limited success for CGRP 8-37 treatment but significant improvement with anti-CGRP Ab. Discussion was also added on future directions to improve global CGRP KO model using a more selective approach.

September 4, 2024

RE: Life Science Alliance Manuscript #LSA-2023-02041-TRR

Dr. Yusuke Shiozawa
Wake Forset University School Of Medicine
Medical Center Blvd
Winston-Salem 27157

Dear Dr. Shiozawa,

Thank you for submitting your revised manuscript entitled "Crosstalk between bone metastatic cancer cells and sensory nerves in bone metastatic progression". We would be happy to publish your paper in Life Science Alliance pending final revisions necessary to meet our formatting guidelines.

Along with points mentioned below, please tend to the following:
-please be sure that the authorship listing and order is correct

FIGURE CHECKS:

-please add sizes next to blots in Figure 8B, 9A, S2E and S3C

A. FINAL FILES:

B. MANUSCRIPT ORGANIZATION AND FORMATTING:

Sincerely,

September 6, 2024

RE: Life Science Alliance Manuscript #LSA-2023-02041-TRRR

Dr. Yusuke Shiozawa
Wake Forset University School Of Medicine
Medical Center Blvd
Winston-Salem 27157

Dear Dr. Shiozawa,

Thank you for submitting your Research Article entitled "Crosstalk between bone metastatic cancer cells and sensory nerves in bone metastatic progression". It is a pleasure to let you know that your manuscript is now accepted for publication in Life Science Alliance. Congratulations on this interesting work.

DISTRIBUTION OF MATERIALS:

Again, congratulations on a very nice paper. I hope you found the review process to be constructive and are pleased with how the manuscript was handled editorially. We look forward to future exciting submissions from your lab.

Sincerely,
